# Probabilistic Analysis of Stable Matching in Large Markets with Siblings

## Abstract

We study a practical matching problem that involves assigning children to daycare centers. The collective preferences of siblings from the same family introduce complementarities, which can lead to the non-existence of stable matchings, as observed in the well-studied hospital-doctor matching problems involving couples. Intriguingly, stable matchings have been observed in real-world daycare markets, even with a substantial number of sibling applicants.

Our research systematically explores the presence of stable matchings in these markets. We conduct a probabilistic analysis of large random matching markets that incorporate sibling preferences. Specifically, we examine scenarios where daycares have similar priorities over children, a common characteristic in practical markets. Our analysis reveals that as the market size approaches infinity, the likelihood of stable matchings existing converges to 1.

To facilitate our investigation, we introduce significant modifications to the Sorted Deferred Acceptance algorithm proposed by Ashlagi et al. [2014]. These adaptations are essential to accommodate a more stringent stability concept, as the original algorithm may yield matchings that fail to meet this criterion. By leveraging our revised algorithm, we successfully identify stable matchings in all real-life datasets examined. Additionally, we conduct comprehensive empirical investigations using synthetic datasets to validate the efficacy of our algorithm in identifying stable matchings.

## 1 Introduction

Stability is a foundational concept in preference-based matching theory [Roth and Sotomayor, 1990], with significant implications for both theoretical frameworks and practical applications [Roth, 2008]. Its importance was underscored by the awarding of the 2012 Nobel Prize in Economics. This fundamental concept is crucial for the success of various markets, including the National Resident Matching Program [Roth, 1984] and public school choice programs [Abdulkadiroğlu and Sönmez, 2003, Abdulkadiroğlu et al., 2005].

Despite its significance, the challenge posed by complementarities in preferences often leads to the absence of a stable matching. A persistent issue in this context is the incorporation of couples into centralized clearing algorithms for professionals like doctors and psychologists [Roth and Peranson, 1999]. Couples typically view pairs of jobs as complements, which can result in the non-existence of a stable matching [Roth, 1984, Klaus and Klijn, 2005]. Moreover, verifying the existence of a stable matching is known to be NP-hard, even in restrictive settings [Ronn, 1990, McDermid and Manlove, 2010, Biró et al., 2014].

Nevertheless, real-life markets of substantial scale do exhibit stable matchings even in the presence of couples. For example, in the psychologists' markets, couples constituted only about $1\%$ of all

participants from 1999 to 2007. Kojima et al. [2013] and Ashlagi et al. [2014] demonstrate that if the proportion of couples grows sufficiently slowly compared to the number of single doctors, then a stable matching is very likely to exist in a large market.

In this paper, we shift our attention to daycare matching markets in Japan, where the issue of waiting children has become one of the most urgent social challenges due to the scarcity of daycare facilities [Kamada and Kojima, 2023]. The daycare matching problem is a natural extension of matching with couples, analogous to hospitals and doctors, with the notable distinction that the number of siblings in each family can exceed two. We are actively collaborating with multiple municipalities, providing advice to design and implement new centralized algorithms tailored to their specific needs.

The objective of this research is to gain a more nuanced understanding of why stable matchings exist in practical daycare markets. Recently, stable matchings have been reported in these markets where optimization approaches are utilized, but the underlying reasons have not been thoroughly examined [Sun et al., 2023, 2024]. Furthermore, theoretical guarantees established in prior research on matching with couples may not readily extend to the daycare market, primarily due to two key factors. Firstly, a distinctive characteristic of Japanese daycare markets is the substantial proportion, approximately 20%, of children with siblings. This stands in contrast to the assumption of near-linear growth of couples in previous research [Ashlagi et al., 2014]. Secondly, we consider a stronger stability concept tailored for daycare markets. Our proposal has been presented to government officials and esteemed economists, who concur that this modification better suits the daycare markets[1].

Our contributions can be summarized as follows:

Firstly, we formalize a large random market that mirrors the characteristics of realistic daycare markets, incorporating family preferences and daycare priorities generated through probability distributions. A significant trait observed in practical markets is the tendency for daycares to exhibit similar priorities over children. Our central result demonstrates that, in such random markets, the probability of a stable matching existing approaches 1 as the market size tends to infinity (Theorem 1). To the best of our knowledge, this is the first work to explain the existence of stable matchings in these practical daycare markets.

Secondly, we modify the Sorted Deferred Acceptance algorithm [Ashlagi et al., 2014] to address our stronger stability concept, as the original algorithm may not produce a matching that satisfies this criterion (Theorem 2). We carefully rectify and extend the algorithm to meet the stronger stability requirement (Theorem 3). Notably, we employ our modified algorithm to successfully identify stable matchings in all encountered real-life datasets. Additionally, we generate a large number of synthetic datasets that closely resemble real-life markets to assess the algorithm's effectiveness across diverse scenarios.

## 2   Related Work

We next provide a brief summary of some papers that are closely related to our work. A more detailed literature review is presented in Appendix A. A classical work on matching with couples, conducted by Kojima et al. [2013], illustrates that as the market size approaches infinity, the probability of a stable matching existing converges to 1, given the growth rate of couples is suitably slow in relation to the market size, e.g., when the number of couples is $\sqrt{n}$ where $n$ represents the number of singles. Ashlagi et al. [2014] propose an improved matching algorithm, building on the foundation laid by Kojima et al. [2013]. This refined algorithm demonstrates that, even if the number of couples grows at a near-linear rate of $n^\epsilon$ with $0 < \epsilon < 1$, a stable matching can still be found with high probability. In contrast, Ashlagi et al. [2014] highlight that as the number of couples increases at a linear rate, the probability of a stable matching existing diminishes significantly. In practical applications, the National Resident Matching Program employed a heuristic based on the incremental algorithm proposed by Roth and Vate [1990]. Biró et al. [2016] proposed a different approach involves the utilization of the Scarf algorithm [Scarf, 1967] to identify a fractional matching. If the outcome proves to be integral, it is then considered a stable matching. Moreover, researchers have explored the application of both integer programming and constraint programming to address the complexities of matching with couples [Manlove et al., 2007, Biró et al., 2014, Manlove et al., 2017]. Notably,

---

[1]To preserve anonymity, their identities are not disclosed in this submission.

these methodologies have recently been adapted in the daycare matching market as well [Sun et al., 2023, 2024].

# 3 Preliminaries

In this section, we present the framework of a daycare market, expanding upon the classical problem of hospital-doctor matching with couples. We also generalize three fundamental properties that have been extensively examined in the literature of two-sided matching markets.

## 3.1 Model

The daycare matching problem is represented by the tuple $I = (C, F, D, Q, \succ_F, \succ_D)$, where $C$, $F$ and $D$ denote sets of children, families, and daycare centers, respectively.

Each child $c \in C$ belongs to a family denoted as $f(c) \in F$. Each family $f \in F$ is associated with a subset of children, denoted as $C(f) \subseteq C$. In cases where a family contains more than one child, e.g., $C(f) = \{c_1, \cdots, c_k\}$ with $k > 1$, these siblings are arranged in a predefined order, such as by age.

Let $D$ represent a set of daycare centers, referred to as "daycares" for brevity. A dummy daycare denoted as $d_0$ is included in $D$, signifying the possibility of a child being unmatched. Each individual daycare $d$ establishes a quota, denoted as $Q(d)$, where the symbol $Q$ represents all quotas.

Each family $f$ reports a strict *preference ordering* $\succ_f$, defined over tuples of daycare centers, reflecting the collective preferences of the children within $C(f)$. The notation $\succ_{f,j}$ is used to represent the $j$-th tuple of daycares in $\succ_f$, and the overall preference profile of all families is denoted as $\succ_F$.

**Example 1.** *Consider family $f$ with $C(f) = \{c_1, c_2, \ldots, c_k\}$ where the children are arranged in a predetermined order. A tuple of daycares in $\succ_f$, denoted as $(d_1^*, d_2^*, \ldots, d_k^*)$, indicates that for each $i \in \{1, 2, \ldots, k\}$, child $c_i$ is matched to some daycare $d_i^* \in D$. It's possible that $d_i^* = d_j^*$, indicating that both child $c_i$ and child $c_j$ are matched to daycare $d_i^*$.*

Each daycare $d \in D$ maintains a strict *priority ordering* $\succ_d$ over $C \cup \emptyset$, encompassing both the set of children $C$ and an empty option. A child $c \in C$ is considered acceptable to daycare $d$ if $c \succ_d \emptyset$, and deemed unacceptable if $\emptyset \succ_d c$. The priority profile of all daycares is denoted as $\succ_D$.

A *matching* $\mu$ is defined as a function $\mu : C \cup D \to C \cup D$ satisfying the following conditions: i) $\forall c \in C$, $\mu(c) \in D$, ii) $\forall d \in D$, $\mu(d) \subseteq C$, and iii) $\forall c \in C$, $\forall d \in D$, $\mu(c) = d$ if and only if $c \in \mu(d)$. Given a matching $\mu$, we designate $\mu(c)$ as the *assignment* of child $c$ and $\mu(d)$ as the assignment of daycare $d$. For a family $f$ with children $C(f) = \{c_1, ..., c_k\}$, we denote the assignment for family $f$ as $\mu(f) = \big(\mu(c_1), ..., \mu(c_k)\big)$.

## 3.2 Fundamental Properties

The first property, individual rationality, stipulates that each family is matched to some tuple of daycares that are weakly better than being unmatched, and no daycare is matched with an unacceptable child. It is noteworthy that each family is considered an agent, rather than individual children.

**Definition 1** (Individual Rationality). *A matching $\mu$ satisfies individual rationality if i) $\forall f \in F, \mu(f) \succ (d_0, \cdots, d_0)$ or $\mu(f) = (d_0, \cdots, d_0)$, and ii) $\forall d \in D, \forall c \in \mu(d), c \succ_d \emptyset$.*

Feasibility in Definition 2 necessitates that i) each child is assigned to one daycare including the dummy daycare $d_0$, and ii) the number of children matched to each daycare $d$ does not exceed its specific quota $Q(d)$.

**Definition 2** (Feasibility). *A matching $\mu$ is* feasible *if it satisfies the following conditions: i) $\forall c \in C$, $|\mu(c)| = 1$, and ii) $\forall d \in D$, $|\mu(d)| \leq Q(d)$.*

Stability is a well-explored solution concept within the domain of two-sided matching theory. Before delving into its definition, we introduce the concept of a *choice function* as outlined in Definition 3. It captures the intricate process by which daycares select children, capable of incorporating various considerations such as priority, diversity goals, and distributional constraints (see, e.g., [Hatfield and Milgrom, 2005, Aziz and Sun, 2021, Suzuki et al., 2023, Kamada and Kojima, 2023]). Following the work by Ashlagi et al. [2014], our choice function operates through a greedy selection of children based on priority only, simplifying the representation of stability.

**Definition 3** (Choice Function of a Daycare). *For a given set of children $C' \subseteq C$, the* choice function *of daycare $d$, denoted as $\mathrm{Ch}_d(C') \subseteq C'$, selects children one by one in descending order of $\succ_d$ without exceeding quota $Q(d)$.*

In this paper, we explore a slightly stronger stability concept than the original one studied in Ashlagi et al. [2014]. It extends the idea of eliminating blocking pairs [Gale and Shapley, 1962] to address the removal of blocking coalitions between families and a selected subset of daycares.

**Definition 4** (Stability). *Given a feasible and individually rational matching $\mu$, family $f$ with children $C(f) = \{c_1, ..., c_k\}$ and the $j$-th tuple of daycares $\succ_{f,j} = (d_1^*, ..., d_k^*)$ in $\succ_f$, form a* blocking coalition *if the following two conditions hold,*
*(1) family $f$ prefers $\succ_{f,j}$ to its current assignment $\mu(f)$, i.e., $(d_1^*, ..., d_k^*) \succ_f \mu(f)$, and*
*(2) for each distinct daycare $d$ in $(d_1^*, ..., d_k^*)$, $C(f, j, d) \subseteq \mathrm{Ch}_d((\mu(d) \setminus C(f)) \cup C(f, j, d))$ holds, where $C(f, j, d) \subseteq C(f)$ denotes a subset of children who apply to daycare $d$ with respect to $\succ_{f,j}$.*

*A feasible and individually rational matching satisfies stability if no blocking coalition exists.*

Consider the input to $\mathrm{Ch}_d(\cdot)$ in Condition 2. First, we calculate $\mu(d) \setminus C(f)$, representing the children matched to $d$ in matching $\mu$ but not from family $f$. Then, we consider $C(f, j, d)$, which denotes the subset of children from family $f$ who apply to $d$ according to the tuple of daycares $\succ_{f,j}$.

This process accounts for situations where a child $c$ is paired with $d$ in $\mu$ but is not included in $C(f, j, d)$, indicating that $c$ is applying to a different daycare $d' \neq d$ according to $\succ_{f,j}$. Consequently, child $c$ has the flexibility to pass his assigned seat from $d$ to his siblings in need. Otherwise, child $c$ would compete with his siblings for seats at $d$ despite he intends to apply elsewhere.

In contrast, the original concept by Ashlagi et al. [2014] does not take siblings' assignments into account. We illustrate the differences between the two concepts in Example 2. More detailed motivation for our definition and further discussions are provided in Appendices B.1 and B.2.

**Example 2** (Example of Stability). *Consider one family $f$ with two children $C(f) = \{c_1, c_2\}$. There are three daycares: $D = \{d_0, d_1, d_2\}$, each with one slot. The preference profile of family $f$ is $(d_1, d_2) \succ_f (d_2, d_0)$. Each daycare prefers $c_1$ over $c_2$.*

*The matching $(d_2, d_0)$ is deemed stable by Ashlagi et al. [2014], but it is not considered stable by Definition 4. This is because it is blocked by family $f$ and the pair $(d_1, d_2)$. Here, child $c_1$ passes his seat at $d_2$ to $c_2$, allowing both children to potentially be matched to a more preferred assignment.*

It is well-known that a stable matching is not guaranteed when couples exist [Roth, 1984]. We provide an example to illustrate that even when each family has at most two children, and all daycares have the same priority ordering over children, a stable matching may not exist. Please refer to Appendix B.3 for details.

# 4 Random Daycare Market

To analyze the likelihood of a stable matching in practice, we proceed to introduce a random market where preferences and priorities are generated from probability distributions. Formally, we represent a random daycare market as $\tilde{I} = (C, F, D, Q, \alpha, \beta, L, \mathcal{P}, \rho, \sigma, \mathcal{D}_{\succ_0, \phi}, \varepsilon)$.

Let $|C| = n$ and $|D| = m$ denote the number of children and daycares, respectively. Throughout this paper, we assume that $m = \Omega(n)$. To facilitate analysis, we partition the set $F$ into two distinct groups labeled $F^S$ and $F^O$, signifying the sets of families with and without siblings, respectively. Correspondingly, $C^S$ and $C^O$ represent the sets of children with and without siblings, respectively. The parameter $\alpha$ signifies the percentage of children with siblings. Then we have $|C^O| = (1 - \alpha)n$ and $|C^S| = \alpha n$. For each family $f$, the size of $C(f)$ is constrained by a constant $\beta$, expressed as $\max_{f \in F} |C(f)| \leq \beta$.

## 4.1 Preferences of Families

We adopt the approach outlined in Kojima et al. [2013] to generate family preferences through a two-step process. In the first step, we independently generate preference orderings for each child from a probability distribution $\mathcal{P}$ on daycares $D$. Subsequently, we employ a function $\rho$ to aggregate the individual preferences of children within each family into a collective preference ordering.

The procedure for generating preference orderings for each child operates as follows. Let $\mathcal{P} = (p_d)_{d \in D}$ be a probability distribution, where $p_d$ represents the probability of selecting daycare $d$. For each child $c$, start with an empty list, independently choose a daycare $d$ from $\mathcal{P}$, and add it to the list if it is not already included. Repeat this process until the length of the list reaches the maximum length $L$, a relatively small constant in practice.

We adhere to the assumption that the distribution $\mathcal{P}$ satisfies a *uniformly bounded* condition, as assumed in the random market by Kojima et al. [2013] and Ashlagi et al. [2014].

**Definition 5** (Uniformly Bounded). *For all $d, d' \in D$, the ratio of probabilities $p_d/p_{d'}$ falls within the interval $[1/\sigma, \sigma]$ with a constant $\sigma \geq 1$.*

**Lemma 1.** *Under the uniformly bounded condition, the probability $p_d$ of selecting any daycare $d$ is limited by $\sigma/m$ where $m$ denotes the total number of daycares.*

For families with multiple siblings, we do not impose additional constraints on the function $\rho$ that aggregates individual preferences into collective preferences.

## 4.2 Priorities of Daycares

A notable departure from previous work [Kojima et al., 2013] and [Ashlagi et al., 2014], is our adoption of the Mallows model [Mallows, 1957] to generate daycare priority orderings over children. In the Mallows model, represented as $\mathcal{D}_{\succ_0, \phi}$, a reference ordering $\succ_0$ is first determined. New orderings are then probabilistically generated based on this reference, controlled by a dispersion parameter $\phi$. This model is widely used for preference generation in diverse contexts [Lu and Boutilier, 2011, Brilliantova and Hosseini, 2022]. Let $S$ denote the set of all orderings over $C$.

**Definition 6** (Kendall-tau Distance). *For a pair of orderings $\succ$ and $\succ'$ in $S$, the Kendall-tau distance, denoted by $\mathrm{inv}(\succ, \succ')$, is a metric that counts the number of pairwise inversions between these two orderings. Formally, $\mathrm{inv}(\succ, \succ') = |\{c, c' \in C \mid c \succ' c' \text{ and } c' \succ c\}|$.*

**Definition 7** (Mallows Model). *Let $\phi \in (0, 1]$ be a dispersion parameter and $Z = \sum_{\succ \in S} \phi^{\mathrm{inv}(\succ, \succ_0)}$. The* Mallows distribution *is a probability distribution over $S$. The probability that an ordering $\succ$ in $S$ is drawn from the Mallows distribution is given by*

$$\Pr[\succ] = \frac{1}{Z} \, \phi^{\mathrm{inv}(\succ, \succ_0)}.$$

The dispersion parameter $\phi$ characterizes the correlation between the sampled ordering and the reference ordering $\succ_0$. Specifically, when $\phi$ is close to 0, the ordering drawn from $\mathcal{D}_{\succ_0, \phi}$ is almost the same as the reference ordering $\succ_0$. On the other hand, when $\phi = 1$, $\mathcal{D}_{\succ_0, \phi}$ corresponds to the uniform distribution over all permutations of $C$.

In the practical daycare matching market, every municipality assigns a unique priority score to each child, establishing a strict priority order utilized and slightly adjusted by all daycares. Siblings within the same family usually share identical priority scores, with ties being resolved arbitrarily.

Motivated by this observation, we construct a reference ordering $\succ_0$ as follows: Begin with an empty list and include all children $C^O$ in the list. For each family $f \in F^S$, add children $C(f)$ to the list with a probability of $1/n^{1+\varepsilon}$, and include $f$ in the list with a probability of $1 - 1/n^{1+\varepsilon}$ for a constant $\varepsilon > 0$. Subsequently, shuffle all permutations of the elements in the list. Finally, $\succ_0$ is drawn from a uniform distribution over all permutations of the shuffled elements in the list. In other words, with a probability of $1/n^{1+\varepsilon}$, we treat siblings from the same family separately, and with a probability of $1 - 1/n^{1+\varepsilon}$, we treat them as a whole, or more precisely, as a continuous block in $\succ_0$.

**Definition 8** (Diameter). *Given a reference ordering $\succ_0$ over children $C$, we define the* diameter *of family $f$, denoted by $\mathrm{diam}_f$, as the greatest difference in $\succ_0$ among $C(f)$. Formally, $\mathrm{diam}_f = \left| \{c \in C \mid \max_{c' \in C(f)} c' \succ_0 c \succ_0 \min_{c'' \in C(f)} c''\} \right| + 2$ where $\max_{c \in C(f)} c$ (resp. $\min_{c \in C(f)} c$) refers to the child in $C(f)$ with the highest (resp. lowest) priority in $\succ_0$.*

The methodology employed to generate the reference ordering $\succ_0$ above adheres to the following condition. For each family $f$ with siblings, we have $\Pr[\mathrm{diam}_f \geq |C(f)|] \leq \frac{1}{n^{1+\varepsilon}}$ from the construction.

We concentrate on a random market $\tilde{I}$ where all parameters are set as mentioned above. Our main result is encapsulated in the following theorem.

**Theorem 1.** *Given a random market $\tilde{I}$ with $\phi = O(\log n/n)$, the probability of the existence of a stable matching converges to $1$ as $n$ approaches infinity.*

We will prove Theorem 1 by demonstrating that an algorithm, namely the Extended Sorted Deferred Acceptance algorithm (to be defined in the next section), produces a stable matching with a probability that converges to 1 in the random market.

# 5 Extended Sorted Deferred Acceptance

In this section, we propose the Extended Sorted Deferred Acceptance (ESDA) algorithm, a heuristic approach that has proven effective in computing stable matchings across a variety of real-life daycare datasets. Importantly, the ESDA algorithm serves as a foundational component in our probability analysis for large random markets.

The ESDA algorithm is an extension of the Sorted Deferred Acceptance (SDA) algorithm [Ashlagi et al., 2014], originally designed for matching with couples. More details of the SDA algorithm are presented in Appendix C.3. In the following theorem, we demonstrate that the SDA algorithm may not produce a stable matching with respect to Definition 4 when it terminates without failure. The proof of Theorem 2 is presented in Appendix C.4.

**Theorem 2.** *The matching returned by the original SDA algorithm may not be stable.*

We next give an informal description of ESDA. The algorithm begins by computing a stable matching without considering families with siblings, denoted as $F^S$, using the Deferred Acceptance algorithm (see Appendix C.1). Subsequently, it sequentially processes each family, denoted as $f$, based on a predefined order denoted as $\pi$. Children without siblings who are displaced by family $f$ are processed individually, enabling them to apply to daycare centers from their top choices in their preference orderings. If any child from family $f' \in F^S$ with siblings is rejected during this process, a new order $\pi'$ is attempted, with $f$ being inserted before $f'$. If the outcome before inserting family $f$ becomes different after processing family $f$, then we check whether family $f$ can be matched to a better tuple of daycares from their top choices. The algorithm terminates and returns a failure if any child from family $f$ is rejected or if the same permutation has been attempted twice.

We offer a brief elucidation on the differences between our ESDA algorithm and the original SDA. Firstly, the input to the choice function of daycares differs. In our algorithm, children have the option to transfer their allocated seats to other siblings, a feature not present in the original SDA. Secondly, we meticulously examine whether any family could establish a blocking coalition with a tuple of daycares that previously rejected it whenever the assignment of any child without siblings is changed. In contrast, SDA goes through each tuple of daycares once without performing this check.

We illustrate how ESDA works through Example 3. A formal description of ESDA is presented in Algorithm 1 in Appendix D, along with all technical details.

**Example 3.** *Consider three families $f_1$ with $C(f_1) = \{c_1, c_2\}$, $f_2$ with $C(f_2) = \{c_3, c_4\}$ and $f_3$ with $C(f_3) = \{c_5, c_6\}$. There are five daycares denoted as $D = \{d_1, d_2, d_3, d_4, d_5\}$, each with one available slot. The order $\pi$ is initialized as $\{1, 2, 3\}$. The preference profile of the families and the priority profile of the daycares are outlined as follows:*

$$\begin{aligned}
&\succ_{f_1}: (d_1, d_2), (d_1, d_4) &&\succ_{d_1}: c_1, c_5 &&\succ_{d_2}: c_6, c_2 \\
&\succ_{f_2}: (d_3, d_4), (d_5, d_4) &&\succ_{d_3}: c_3, c_5 &&\succ_{d_4}: c_6, c_4, c_2 \\
&\succ_{f_3}: (d_1, d_4), (d_3, d_4), (d_5, d_2) &&\succ_{d_5}: c_3, c_5
\end{aligned}$$

***Iteration 1:*** *With order $\pi_1 = \{1, 2, 3\}$, family $f_1$ secured a match by applying to daycares $(d_1, d_2)$, followed by family $f_2$ obtaining a match with applications to $(d_3, d_4)$. However, family $f_3$ faced rejections at $(d_1, d_4)$ and $(d_3, d_4)$ before successfully securing acceptance at $(d_5, d_2)$, leading to the displacement of family $f_1$. Thus we generate a new order $\pi_2 = \{3, 1, 2\}$ by inserting 3 before 1.*

***Iteration 2:*** *With order $\pi_2 = \{3, 1, 2\}$, family $f_3$ secures a match at $(d_1, d_4)$. Then family $f_1$ applies to $(d_1, d_2)$ and also secures a match, resulting in the eviction of family $f_3$. This leads to the generation of a modified order $\pi_3 = \{1, 3, 2\}$ with 1 preceding 3.*

***Iteration 3:*** *With order $\pi_3 = \{1, 3, 2\}$, family $f_1$ secures a match at $(d_1, d_2)$. Subsequent applications by $f_3$ result in a match at $(d_3, d_4)$, but $f_2$ remains unmatched due to rejections at $(d_3, d_4)$ and $(d_5, d_4)$.*

*The algorithm terminates, returning a stable matching $\mu$ with $f_1$ matched to $(d_1, d_2)$ and $f_3$ matched to $(d_3, d_4)$, while $f_2$ remains unmatched.*

## 5.1 Termination without Failure

We demonstrate that ESDA always generates a stable matching if it does not terminate with failures. Our proof relies on the following two facts, which are formally presented in Appendix D.1. First, we establish that the number of matched children at each daycare does not decrease as long as no family in $F^S$ is rejected and no child passes their seat to other siblings during the execution of ESDA. Second, we prove that for a given order $\pi$ over $F^S$, if the rank of the matched child at any daycare increases, then ESDA cannot produce a matching with respect to $\pi$. The detailed proof for Theorem 3 is presented in Appendixes D.1 and D.2.

**Theorem 3.** *Given an instance of $I$, if ESDA returns a matching without failure, then the yielded matching is stable. In addition, ESDA always terminates in a finite time, either returning a matching or a failure.*

## 5.2 Two Types of Failure of ESDA

Theorem 3 states that if the algorithm successfully concludes, it results in a stable matching. Conversely, the algorithm returns failures in two scenarios, suggesting that a stable matching may not exist, even if one indeed exists.

Formally, a *Type-1 Failure* happens when, during the insertion of a family $f \in F^S$, a child $c \in C(f)$ initiates a rejection chain that ends with another child $c' \in C(f)$ from the same family, where all other children in the chain do not have siblings. This failure is further divided into two cases based on whether $c = c'$ holds: Type-1-a Failure when $c = c'$ and Type-1-b Failure when $c \neq c'$.

A *Type-2 Failure* occurs if there exist two families $f_1, f_2 \in F^S$ satisfying the following conditions: i) $f_1$ appears before $f_2$ in the current order $\pi$, ii) There exists a rejection chain starting from $f_2$ and ending with $f_1$ where all other families in the chain have an only child, and iii) A new order $\pi'$, obtained by placing $f_2$ in front of $f_1$, has been attempted and stored in the set of $\Pi$, which keeps track of permutations tried during the ESDA process.

These two types of failures are crucial when analyzing the probability of the existence of stable matchings in a large random market. Detailed examples illustrating these two types of failures can be found in Appendix D.3.

# 6 Skecthed Proof of Theorem 1

Our main approach to proving Theorem 1 is to set an upper limit on the likelihood of encountering the two types of failure in the ESDA algorithm.

The following two lemmas establish that as $n$ approaches infinity, Type-1-a and Type-1-b Failures are highly unlikely to occur when the dispersion parameter $\phi$ is on the order of $O(\log n / n)$. We defer the proofs for these results to Appendices E.2 and E.3, respectively.

**Lemma 2.** *Given a random market $\tilde{I}$ with $\phi = O(\log n / n)$, the probability of Type-$1$-a Failure in the SDA algorithm is bounded by $O\big((\log n)^2 / n\big)$.*

**Lemma 3.** *Given a random market $\tilde{I}$ with $\phi = O(\log n / n)$, the probability of Type-$1$-b Failure in the SDA algorithm is bounded by $O\big((\log n)^2 / n\big) + O(n^{-\varepsilon})$.*

We introduce concepts of *domination* and *nesting* to analyze the case of Type-2 Failure.

**Definition 9** (Domination)**.** *Given a priority ordering $\succ$, we say that family $f$ dominates $f'$ w.r.t. $\succ$ if $\max_{c \in C(f)} c \succ \min_{c' \in C(f')} c'$ where $\max_{c \in C(f)} c$ (resp. $\min_{c \in C(f)} c$) represents the child in $C(f)$ with the highest (resp. lowest) priority under the priority ordering $\succ$.*

In simple terms, if $f$ dominates $f'$, then there is a possibility that $f'$ will be rejected by daycares with a certain order $\succ$ due to an application of $f$.

Intuitively, a Type-2 Failure can arise from a cycle in which two families with siblings reject each other. We introduce the concept of *nesting* as follows.

**Definition 10** (Nesting). *Given a priority ordering $\succ$, two families $f$ and $f'$ are said to be* nesting *if they mutually dominate each other under $\succ$.*

**Example 4.** *Consider three families $F = \{f_1, f_2, f_3\}$, each with two children: $C(f_1) = \{c_1, c_2\}$, $C(f_2) = \{c_3, c_4\}$, and $C(f_3) = \{c_5, c_6\}$. Suppose there is a priority ordering $\succ$: $c_1$, $c_3$, $c_5$, $c_2$, $c_4$, $c_6$. In this case, all pairs in $F$ nest with each other with respect to $\succ$.*

We next show that if any two families do not nest with each other with respect to $\succ_0$, then Type-2 Failure is unlikely to occur as $n$ tends to infinity in Lemma 4. We defer the proof to Appendix E.4.

**Lemma 4.** *Given a random market $\tilde{I}$ with $\phi = O(\log n/n)$, and for any two families $f, f' \in F^S$ that are not nesting with each other with respect to $\succ_0$, then Type-2 Failure occurs with a probability of at most $O(\log n/n)$.*

Following an analysis of the probability that any two pairs of families from $F^S$ nest with each other with respect to the reference ordering $\succ_0$, we establish the probability of Type-2 Failure in Lemma 5.

**Lemma 5.** *Given a random market $\tilde{I}$ with $\phi = O(\log n/n)$, the probability of Type-2 Failure occurring is bounded by $O(\log n/n) + O(n^{-2\varepsilon})$.*

Lemma 2, Lemma 3, and Lemma 5 imply the existence of a stable matching with high probability for the large random market, thus concluding the proof of Theorem 1. Further elaboration and details can be found in Appendix E.

# 7 Experiments

In this section, we conduct comprehensive experiments to evaluate the effectiveness of our proposed ESDA algorithm. The experimental results demonstrate our hypothesis that a stable matching exists with high probability when daycare centers have similar priority orderings over children.

We analyze two types of datasets. Firstly, we evaluate our algorithm using six real-life datasets provided by three municipalities. In Appendix F.2, we provide a detailed description of the practical daycare matching markets based on datasets. In addition, we introduce slight modifications to daycare priorities while keeping other factors constant. Secondly, we generate synthetic datasets that mirror the characteristics of real-life markets but on a much larger scale. By adjusting the dispersion parameter in the Mallows model, we create daycare priorities with varying degrees of similarity.

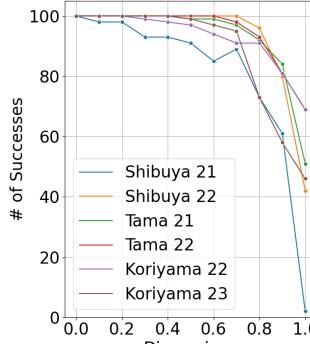

Figure 1: Results of experiments on real-world data perturbed by the Mallows distributions.

Given the limitations of the ESDA algorithm in computing stable matchings in certain scenarios, we employ a constraint programming (CP) approach as an alternative. This method consistently generates a stable matching whenever one exists [Sun et al., 2024]. We implement them in Python and execute them on a standard laptop without additional computational resources. To generate priorities from the Mallows distributions, we utilize the PrefLib library [Mattei and Walsh, 2013]

## 7.1 Experiments on Real-life Datasets

We present the experimental results on the six real-life datasets. It is noteworthy that the ESDA algorithm not only successfully identifies a stable matching but also consistently produces the same outcome as the constraint programming (CP) solution for all datasets. Moreover, the ESDA algorithm achieves a computation time that is more than 10 times faster than the CP (see Table 5 in Appendix F.2).

To investigate the importance of similarity in daycare priorities on the performance of ESDA, we generate new datasets by perturbing the original real-world data using Mallows distributions. For each daycare, we independently sample priority orders from the Mallows distribution with varying

dispersion parameters and replace the original priority order. We consider dispersion parameters ranging from 0.0 to 1.0 in increments of 0.1 and conduct 100 experiments for each case. Figure 1 illustrates the results, demonstrating that ESDA successfully computes a stable matching in more than 80% of cases when the dispersion parameter $\phi$ is at most 0.8. It is worth noting that when $\phi = 0.0$, daycare priorities are identical to the original priorities. However, when the dispersion parameter is large, the ESDA may only find a stable matching in less than 50% of cases, even if one may exist.

## 7.2 Experiments on Synthetic Datasets

We illustrate the steps to generate synthetic datasets. Initially, we define the number of families, denoted by $|F|$, drawn from the set $\{500, 1000, 2000, 3000, 5000, 10000\}$. We next fix the parameter $\alpha$, representing the percentage of children with siblings $C^S$, as $\alpha = 0.2$. For families with siblings, denoted as $F^S$, 80% of them consist of two children each, while the remaining 20% have three children each. The number of daycares, denoted by $|D|$, is set to $0.1 * |F|$. For each child $c$ without siblings in $C^O$, we randomly select 5 daycares from $D$. For each family $f$ in $F^S$ with siblings, we generate an individual preference ordering of length 10 uniformly from $D$ for each child $c \in C(f)$ and create all possible combinations. Finally, we uniformly choose a joint preference ordering of length 10. The dispersion parameter $\phi$ varies within the range $\{0.0, 0.3, 0.5\}$, while the parameter $\varepsilon$ used to generate common priorities $\succ_0$ remains fixed at 1. For each specified setting, we generate 10 instances. The figures in the first row show the number of successful runs out of the 10 experiments. In the second row, we report the mean computational complexity along with its 95% confidence intervals, calculated only for the instances where the algorithm successfully found a stable matching.

Regarding the experimental findings, the ESDA algorithm consistently identified a stable matching in all experiments. In addition to stability analysis, we conducted a comparison of the running time between the ESDA algorithm and the CP algorithm. Despite the potential requirement for the ESDA algorithm to check all permutations of $F^S$ in the worst case scenario, it consistently demonstrated notably faster performance than the CP algorithm across all cases.

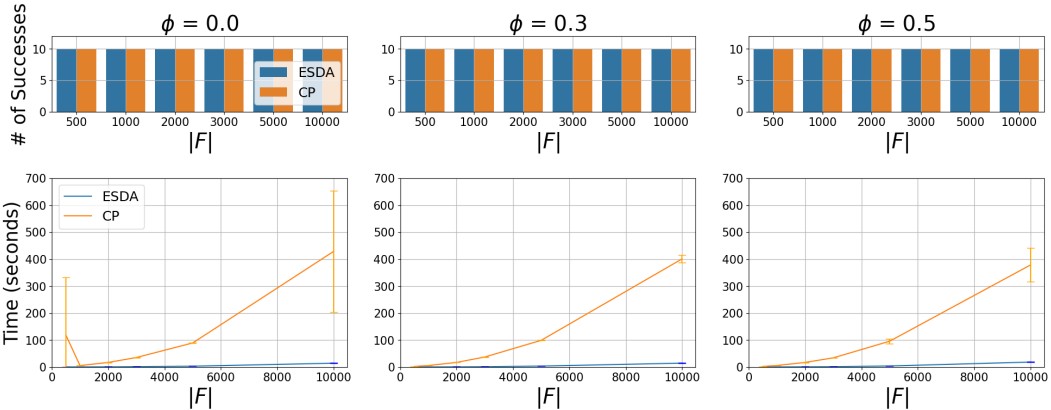

Figure 2: Results of experiments on synthetic data.

## 8 Conclusion

In this study, we investigate the factors contributing to the existence of stable matching in practical daycare markets. We identify the shared priority ordering among all daycares as one of the primary reasons. Our contribution includes a probability analysis for such large random markets and the introduction of the ESDA algorithm to identify stable matchings in practical datasets. Experimental results demonstrate the utility of ESDA under various conditions, suggesting its potential scalability to larger markets where optimization solutions, such as integer programming or constraint programming, may exhibit much longer processing times.

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

# A   Related Work

Ronn [1990] initially established that verifying the existence of stable matchings in the presence of couples is an NP-hard problem, even if each hospital offers only one position. Follow-up work by McDermid and Manlove [2010] showed this computational intractability result still holds even when couples' preferences are limited to pairs of positions within the same hospital. Furthermore, Biró et al. [2011] demonstrated that it remains NP-hard when all doctors are ranked according to a common order adopted by all hospitals.

A classical work on matching with couples, conducted by Kojima et al. [2013], illustrates that as the market size approaches infinity, the probability of a stable matching existing converges to 1, given the growth rate of couples is suitably slow in relation to the market size, e.g., when the number of couples is $\sqrt{n}$ where $n$ represents the number of singles. Ashlagi et al. [2014] propose an improved matching algorithm, building on the foundation laid by Kojima et al. [2013]. This refined algorithm demonstrates that, even if the number of couples grows at a near-linear rate of $n^\epsilon$ with $0 < \epsilon < 1$, a stable matching can still be found with high probability. In contrast, Ashlagi et al. [2014] highlight that as the number of couples increases at a linear rate, the probability of a stable matching existing diminishes significantly.

Kojima et al. [2013] devised the Sequential Couples Algorithm to address matching problems involving couples, which follows a three-step procedure. First, it computes a stable matching without considering couples, using the DA algorithm. Next, it handles each couple according to a predefined order denoted as $\pi$. Single doctors displaced by couples are accommodated one by one, allowing them to apply to hospitals based on their preferences. However, if an application is made to a hospital where any member of a couple has previously submitted an application, the algorithm declares a failure and terminates, even though a stable matching may indeed exist.

The Sorted Deferred Acceptance (SDA) algorithm, as introduced by Ashlagi et al. [2014], follows a similar trajectory to the Sequential Couples Algorithm. We extend its application to the context of daycare matching with siblings. The algorithm begins by computing a stable matching without considering families with siblings, denoted as $F^S$, using the DA algorithm. Subsequently, it sequentially processes each family, denoted as $f$, based on a predefined order denoted as $\pi$. Children without siblings who are displaced by family $f$ are processed individually, enabling them to apply to daycare centers according to their preferences. If any child from family $f' \in F^S$ with siblings is affected during this process, a new order $\pi'$ is attempted, with $f$ being inserted before $f'$. The algorithm terminates and returns a failure if any child from family $f$ is affected or if the same permutation has been attempted twice.

One potential solution to overcome the non-existence of stable matchings is to explore restricted preference domains. In this regard, Klaus and Klijn [2005] investigated a restricted preference domain known as weak responsiveness, ensuring the presence of stable matchings in the presence of couples. Hatfield and Kojima [2010] introduced the concept of "bilateral substitute" within the framework of matching with contracts [Hatfield and Milgrom, 2005], encompassing matching with couples as a specific case, and they demonstrated that weak responsiveness implies bilateral substitutes.

In practical applications, the National Resident Matching Program employed a heuristic based on the incremental algorithm proposed by Roth and Vate [1990]. Biró et al. [2016] proposed a different approach involves the utilization of the Scarf algorithm [Scarf, 1967] to identify a fractional matching. If the outcome proves to be integral, it is then considered a stable matching. Moreover, researchers have explored the application of both integer programming and constraint programming to address the complexities of matching with couples [Manlove et al., 2007, Biró et al., 2014, Manlove et al., 2017]. Notably, these methodologies have recently been adapted in the daycare matching market as well [Sun et al., 2023, 2024].

Another trend in the literature explores the combination of bandit algorithms with matching market design. In these studies, preferences are initially unknown and are learned through the interactions between the two sides of agents (see [Das and Kamenica, 2005, Liu et al., 2020, Basu et al., 2021, Liu et al., 2021, Jagadeesan et al., 2021, Kong et al., 2022]). This contrasts with our setting, where preferences and priorities are submitted to the system in advance.

## B  Discussion on Stability

### B.1  Motivation

The primary reason for modifying the stability concept lies in the differing selection criteria between hospital-doctor matching and daycare allocation. In the hospital-doctor matching problem, hospitals have preferences over doctors. In contrast, daycare centers use priority orderings based on priority scores to determine which child should be given higher precedence. The priority scoring system is designed to eliminate justified envy and achieve a fair outcome, treating daycare slots as resources to be allocated equitably.

Additionally, it is crucial that siblings do not envy each other, especially when they are not enrolled in the same daycare. Allowing children to transfer their seats to other siblings can potentially reduce waste and increase overall welfare.

We presented this new stability concept to multiple government officials from different municipalities and several renowned economists. They all agreed that the modification is more appropriate for the daycare matching setting.

### B.2  ABH-Stability

The stability concept studied in [Ashlagi et al., 2014] was originally designed for matching with couples and defined by enumerating all possible scenarios. To distinguish it from our concept, we refer to their stability as ABH-stability, named after the authors' initials.

In Definition 11, we consolidate these scenarios into a concise format, which highlights the differences from our definition. The primary distinction from Definition 4 lies in the input to $\mathrm{Ch}_d(\cdot)$ in condition 2: it uses $\mathrm{Ch}_d(\mu(d) \cup C(f, j, d))$, instead of $\mathrm{Ch}_d(\mu(d) \setminus C(f) \cup C(f, j, d))$.

**Definition 11** (ABH-Stability). *Given a feasible and individually rational matching $\mu$, family $f$ with children $C(f) = \{c_1, ..., c_k\}$ and the $j$-th tuple of daycares $\succ_{f,j} = (d_1^*, ..., d_k^*)$ in $\succ_f$, form a blocking coalition if the following two conditions hold,*
*(1) $(d_1^*, ..., d_k^*) \succ_f \mu(f)$, and*
*(2) for each distinct daycare $d$ included in $(d_1^*, ..., d_k^*)$, $C(f, j, d) \subseteq \mathrm{Ch}_d(\mu(d) \cup C(f, j, d))$, where $C(f, j, d)$ denotes a subset of $f$'s children who apply to daycare $d$ with respect to $\succ_{f,j}$.*

*A feasible and individually rational matching satisfies ABH-stability if no blocking coalition exists.*

ABH-Stability maintains alignment with the stability notion presented by Kojima et al. [2013]. In the latter study, the authors explore a responsive preference domain in which daycare priorities are defined over sets of children. Despite differences in the choice function employed, the foundational idea of defining stability exhibits conceptual coherence between these two works.

### B.3  Non-existence of Stable Matchings

**Example 5** (Non-existence of Stable Matchings). *Consider three families: $f_1$ with children $C(f_1) = \{c_1, c_2\}$, $f_2$ with children $C(f_2) = \{c_3, c_4\}$, and $f_3$ with children $C(f_3) = \{c_5, c_6\}$. There are three daycares: $D = \{d_1, d_2, d_3\}$, each with a single slot. The preference profile of the families and the priority profile of the daycares are as follows:*

$$\succ_{f_1}: (d_1, d_2) \quad \succ_{f_2}: (d_2, d_3) \quad \succ_{f_3}: (d_3, d_1)$$
$$\succ_d: c_1, c_6, c_3, c_2, c_5, c_4 \ \ \forall d \in D$$

*We denote the option of being unmatched as $\emptyset$ for brevity. There are three feasible matchings except for the empty matching which can not be stable, namely:*

- *Matching $\mu_1$ where $\mu_1(f_1) = (d_1, d_2)$, $\mu_1(f_2) = (\emptyset, \emptyset)$, and $\mu_1(f_3) = (\emptyset, \emptyset)$.*

- *Matching $\mu_2$ where $\mu_2(f_1) = (\emptyset, \emptyset)$, $\mu_2(f_2) = (d_2, d_3)$, and $\mu_2(f_3) = (\emptyset, \emptyset)$.*

- *Matching $\mu_3$ where $\mu_3(f_1) = (\emptyset, \emptyset)$, $\mu_3(f_2) = (\emptyset, \emptyset)$, and $\mu_3(f_3) = (d_3, d_1)$.*

*Matching $\mu_1$ cannot be stable, because family $f_2$ could form a blocking coalition with a pair of daycares $(d_2, d_3)$, where $\mathrm{Ch}_{d_2}(\{c_2, c_3\}) = \{c_3\}$ and $\mathrm{Ch}_{d_3}(\{c_4\}) = \{c_4\}$. Similarly, matching $\mu_2$ is*

 *blocked by family $f_3$ and daycares $(d_3, d_1)$, and matching $\mu_3$ is blocked by family $f_1$ and daycares*
*$(d_1, d_2)$. Consequencely, none of the matchings $\mu_1$, $\mu_2$, and $\mu_3$ is stable.*

## C   Previous Algorithms

### C.1   Deferred Acceptance (DA)

The Deferred Acceptance (DA) algorithm is a classical algorithm in matching theory under pref-
erences [Gale and Shapley, 1962, Roth, 1985]. The (children-proposing) DA algorithm proceeds
iteratively through the following two phases. In the application phase, children first apply to their
most preferred daycares that have not rejected them so far. In the selection phase, each daycare
selects children based on its priorities from the pool of new applicants in the current round and
the temporarily matched children from the previous round without exceeding specific quotas. The
algorithm terminates when no child submits any further applications. An essential property of the DA
algorithm is that it always converges to a stable matching within polynomial time when siblings are
not involved.

**Definition 12** (Rejection Chain)**.** *When a child $c_1^*$ applies to a daycare $d_1^*$ that is already at full*
*capacity, daycare $d_1^*$ must reject some child $c_2^*$ (which could be $c_1^*$). The rejected child $c_2^*$ then applies*
*to the next available daycare $d_2^*$. If daycare $d_2^*$ is also full, another child $c_3^*$ must be rejected by $d_2^*$*
*and apply to the subsequent daycare $d_3^*$. This sequence continues, forming a rejection chain denoted*
*as $c_1^* \to c_2^* \cdots \to c_t^*$, where $t$ represents the length of the chain.*

*Similarly, rejection chains of families can be defined in the same manner by substituting $c_i^*$ with $f_i^*$,*
*where $c_i^* \in C(f_i^*)$.*

**Definition 13** (Rejection Cycle)**.** *A rejection chain, represented as $c_1^* \to c_2^* \cdots \to c_t^*$, is termed a*
rejection cycle *if it satisfies two additional conditions: i) at least one child in the chain is different*
*from $c_1^*$, i.e., there exists $c' \in \{c_1^*, c_2^*, \cdots, c_t^*\}$ such that $c' \neq c_1^*$, and ii) the rejection chain forms a*
*cycle, commencing and concluding with $c_1^*$, i.e., $c_1^* = c_t^*$.*

*In the case of a rejection cycle involving families, we mandate that i) at least two distinct families*
*are present in the rejection chain, and ii) the rejection chain initiates and concludes with the same*
*family. It is possible that the starting child $c_1^*$ and the ending child $c_t^*$ are different, but they are from*
*the same family.*

In cases where no child has siblings, rejection cycles may occur, but they are guaranteed to eventually
terminate. This termination is ensured by the following reasons: i) When a daycare reaches its quota,
the number of matched children remains constant, even though the set of matched children may
vary. ii) Children cannot be matched to a daycare that previously rejected them, as a daycare never
regrets rejecting a child with lower priority than its currently matched children when it meets its
quota. Consequently, a child does not need to reapply to any daycare that has rejected them.

However, these arguments do no longer hold in the presence of siblings. This is because when one
child is rejected by a daycare, their sibling may be compelled to leave the matched daycare, due to
their joint preferences over tuples of daycares, rather than a rejection. Consequently, vacancies arise
at a daycare that was previously full, enabling a previously rejected child to reapply. This suggests
that a rejection cycle may persist indefinitely.

### C.2   Sequential Couples

The Sequential Couples algorithm, devised by Kojima et al. [2013] to address matching problems
involving couples, follows a three-step procedure. First, it computes a stable matching without
considering couples, using the DA algorithm. Next, it handles each couple according to a predefined
order denoted as $\pi$. Single doctors displaced by couples are accommodated one by one, allowing
them to apply to hospitals based on their preferences. However, if an application is made to a hospital
where any member of a couple has previously submitted an application, the algorithm declares a
failure and terminates, even if a stable matching indeed exists.

## C.3 Sorted Deferred Acceptance

The Sorted Deferred Acceptance (SDA) algorithm, as introduced by Ashlagi et al. [2014], follows a similar trajectory to the Sequential Couples algorithm. We extend its application to the context of daycare matching with siblings. The algorithm begins by computing a stable matching without considering families with siblings, denoted as $F^S$, using the DA algorithm. Subsequently, it sequentially processes each family, denoted as $f$, based on a predefined order denoted as $\pi$. Children without siblings who are displaced by family $f$ are processed individually, enabling them to apply to daycare centers according to their preferences. If any child from family $f' \in F^S$ with siblings is affected during this process, a new order $\pi'$ is attempted, with $f$ being inserted before $f'$. The algorithm terminates and returns a failure if any child from family $f$ is affected or if the same permutation has been attempted twice.

## C.4 Proof of Theorem 2

**Theorem 2.** *The matching returned by the original SDA algorithm may not be stable.*

*Proof.* We present a counterexample in Example 6 to prove Theorem 2.

**Example 6.** *Consider two families: $f_1$ with children $C(f_1) = \{c_1, c_2\}$, $f_2$ with children $C(f_2) = \{c_3\}$. There are three daycares: $D = \{d_1, d_2, d_3\}$, each with a single slot. The preference profile of the families and the priority profile of the daycares are as follows:*

$$\succ_{f_1}: (d_1, d_2), (d_2, d_3), \quad \succ_{f_2}: d_2$$
$$\succ_d: c_1, c_3, c_2 \quad \forall d \in D.$$

*Then, SDA produces a matching $\mu(f_1) = \{(d_2, d_3)\}$ while leaving child $c_3$ unmatched. However, by Definition 4, this matching is not stable. This is because family $f_1$ could form a blocking coalition with $(d_1, d_2)$ by allowing $c_1$ to transfer his seat at $d_2$ to sibling $c_2$.*

This completes the proof of Theorem 2. Note that no matching for this example satisfies stability in Definition 4. ◻

## D  Formal Description of ESDA

The ESDA algorithm commences with the application of the Deferred Acceptance (DA) algorithm to families without siblings $F^O$. The resulting matching is denoted as $\mu^O$. The ESDA algorithm operates with an order $\pi$ defined over the set $\{1, \cdots, |F^S|\}$. To keep track of attempted permutations, we introduce the collection $\Pi$, initialized with $\{\pi\}$.

The pivotal step in the ESDA algorithm involves the sequential insertion of families $F^S$ based on the order $\pi$. Let $\pi(i)$ denote the $i$-th element in $\pi$, starting with $i = 1$, and let $F^S_{\pi(i)}$ denote the $\pi(i)$-th family in $F^S$. We define $\mu$ as the current matching during the ESDA process, and $\mu^i$ denotes the matching before processing the $\pi(i)$-th family in $F^S$. Both $\mu$ and $\mu^i$ are initialized with $\mu^O$.

Consider the $\pi(i)$-th family $f \in F^S$, denoted as $f = F^S_{\pi(i)}$. Family $f$ makes proposals to the $j$-th tuple of daycares, denoted as $\succ_{f,j}$, with the initialization of $j$ at 1. Define $D(f, j)$ as the set of distinct daycares in $\succ_{f,j}$. For each daycare $d \in D(f, j)$, we calculate $C(f, j, d)$, representing the set of children from family $f$ applying to daycare $d$ w.r.t. $\succ_{f,j}$.

According to the choice function outlined in Definition 3, the input is $\mu(d) \setminus C(f) \cup C(f, j, d)$, excluding siblings from $C(f)$ who do not apply to daycare $d$ w.r.t. $\succ_{f,j}$. If $C(f, j, d)$ cannot be chosen by all $d \in D(f, j)$, the algorithm advances to the next tuple of daycares by updating $j \leftarrow j + 1$. Otherwise, family $f$ can be matched to $\succ_{f,j}$ in $\mu$.

Let $A$ denote a set of children who i) do not belong to family $f$ and ii) are involved in the rejection chains when matching $f$ to $\succ_{f,j}$. Two possibilities can arise.
Case 1) If any child from family $f' \in F^S \setminus \{f\}$ is involved in $A$, i.e., $A \cap C(f') \neq \emptyset$, a new order $\pi'$ is generated by inserting $f$ before $f'$. If $\pi'$ has been attempted previously, the algorithm terminates, returning failure (Type 2), a concept that will be detailed shortly. Otherwise, the algorithm restarts with the new order $\pi'$ and add $\pi'$ to $\Pi$.

Case 2) If only children without siblings are involved in $A$, then match $f$ with $\succ_{f,j}$ and leave each child in $A$ unmatched. Let $B$ denote the set of children in $C^O$ who are matched differently under $\mu^i$ (the matching before processing family $f$) and $\mu$ (the current matching). Create a temporary matching $\mu^T \leftarrow \mu$, which is used to verify whether $\mu$ will be modified later. Then the algorithm proceeds to stabilize children in $B$. Select one child, denoted as $b \in B$, and let him apply to a daycare denoted as $x \leftarrow \succ_{f(b),h}$ starting with $h = 1$. If any child from $C(f)$ is rejected during this process, the algorithm terminates, returning failure (Type 1). If any child from family $f' \in F^S \setminus f$ is rejected, a new order is generated following the process described in Case 1). If child $b$ is rejected by daycare $x$, the algorithm explores his next preferred daycare with $h \leftarrow h + 1$, if available. If child $b$ is chosen, then match $b$ to $x$ in $\mu$ and remove $b$ from $B$. Subsequently, if there is a rejected child, it is added to $B$, and the algorithm proceeds to the next child in $B$.

Once $B$ becomes empty, we verify whether $\mu^T$ equals $\mu$. If they are not identical, we revisit family $f$ by setting $i \leftarrow i$; otherwise, we update $\mu^{i+1} \leftarrow \mu$ and proceed to the next family in $F^S$ by setting $i \leftarrow i + 1$.

### D.1 Two Lemmas for Proving Theorem 3

Our proof that ESDA always generates a stable matching if it does not terminate with failures, relies on the following two lemmas. First, we establish that the number of matched children at each daycare does not decrease as long as no family in $F^S$ is rejected and no child passes their seat to other siblings during the execution of ESDA. Then, we prove that for a given order $\pi$ over $F^S$, if the rank of the matched child at any daycare increases, then ESDA cannot produce a matching with respect to $\pi$.

**Lemma 6.** *For a given order $\pi$ over families $F^S$, let $\mu^i(\pi)$ denote the matching obtained during the ESDA procedure before processing family $F^S_{\pi(i)} \in F^S$. The number of matched children at any daycare $d$ does not decrease under matching $\mu^{i+1}(\pi)$ if the following three conditions hold: i) The algorithm does not encounter any type of failure. ii) The order $\pi$ remains unchanged. iii) No child from family $F^S_{\pi(i+1)}$ transfers their seat to other siblings during the ESDA process.*

*Proof.* If the first two conditions hold, then no child from any family $f \in F^S$ is rejected when inserting family $F^S_{\pi(i+1)}$. Consequently, only children without siblings are involved in rejection chains, and each time one child is replaced by another one with a higher daycare priority when the capacity is reached.

Let $f = F^S_{\pi(i+1)}$. If the third condition holds, when family $f$ applies to any tuple of daycares $\succ_{f,j}$, the input to the choice function $\text{Ch}_d(\cdot)$ can be simplified as $\text{Ch}_d(\mu(d) \cup C(f, j, d))$, as no child $c \in C(f)$ passes their seat to other siblings. After the stabilization step, if $f$ reapplies to any tuple $\succ_{f,k}$ that is better than $\mu(f)$, then $f$ is still rejected as each matched child at $d \in D(f, j)$ has a weakly higher priority. Thus, $f$ cannot create new vacancies by moving to a better tuple of daycares. Consequently, the number of matched children at each daycare does not decrease. $\square$

For a given matching $\mu$ and a daycare $d$, let $L(\mu, d)$ represent the rank of the matched child with the lowest priority at daycare $d$, where $1$ denotes the highest priority. Imagine that all vacant slots at each daycare are initially occupied by dummy children assigned the rank $|C| + 1$. As the ESDA algorithm progresses, these dummy children are gradually rejected and replaced by children with higher priorities, resulting in a decrease in $L(\cdot)$.

We will now demonstrate the following lemma.

**Lemma 7.** *Given an order $\pi$ over families $F^S$, if, during the ESDA process, the rank $L(\mu, d)$ increases for any daycare $d$, then ESDA fails to generate a matching under the current order $\pi$ over families $F^S$.*

*Proof.* We next prove Lemma 7 by examining the changes in $L(\mu, d)$ at each daycare $d$ throughout the execution of the ESDA algorithm under a given order $\pi$.

**[Line 1]** The ESDA algorithm begins by employing the DA algorithm on families $F^O$. At each step of the DA algorithm, a rejected child is substituted by another child with a higher priority. Consequently, for each daycare $d$, the value of $L(\mu, d)$ either decreases or remains unchanged.

**Algorithm 1** Extended Sorted Deferred Acceptance (ESDA)

---

**Input:** an instance $I = (C, F, D, Q, \succ_F, \succ_D)$ and a default order $\pi = 1, 2, ..., |F^S|$
**Output:** a stable matching or a failure

1: Apply DA to $F^O$ and denote the obtained matching as $\mu^O$
2: Initialize $\Pi \leftarrow \{\pi\}$, storing the permutations of $\pi$ that have been attempted
3: Initialize $i \leftarrow 1$ with $\pi(i)$ being the $i$-th element in $\pi$
4: Initialize $\mu \leftarrow \mu^O$ (current matching) and $\mu^i \leftarrow \mu^O$ (the matching before processing the $\pi(i)$-th family in $F^S$)
5: **while** $i \leq |F^S|$ **do** {Iterate through $F^S$ according to $\pi$}
6:      Let $f = F^S_{\pi(i)}$ be the $\pi(i)$-th family in $F^S$
7:      Initialize $j \leftarrow 1$
8:      **while** $j \leq |\succ_f|$ **do** {$f$ proposes to $\succ_{f,j}$}
9:          Compute $D(f, j)$, the set of distinct daycares w.r.t. $\succ_{f,j}$
10:         For each $d \in D(f, j)$, compute $C(f, j, d)$, the set of children from family $f$ who apply to $d$ w.r.t. $\succ_{f,j}$
11:         **if** $\exists d \in D(f, j)$ s.t. $C(f, j, d) \not\subseteq \mathrm{Ch}_d\big(\mu(d) \setminus C(f) \cup C(f, j, d)\big)$ **then** {$f$ cannot be matched to $\succ_{f,j}$}
12:            $j \leftarrow j + 1$ {Consider the next tuple of daycares in $\succ_f$}
13:         **else** {$f$ can be matched to $\succ_{f,j}$}
14:            $A \leftarrow \bigcup_{d \in D(f,j)} \big(\mu(d) \setminus \mathrm{Ch}_d(\mu(d) \setminus C(f) \cup C(f, j, d))\big) \setminus C(f)$ {Rejected children from families $F \setminus \{f\}$}
15:            **if** $\exists f' \in F^S \setminus \{f\}$ s.t. $C(f') \cap A \neq \emptyset$ **then** {some child from $f' \in F^S \setminus \{f\}$ is rejected}
16:               Create a new order $\pi'$ by inserting $f$ prior to $f'$.
17:               **if** $\pi' \in \Pi$ **then**
18:                  **return** Failure (Type-2).
19:               **else**
20:                  $\Pi \leftarrow \Pi \cup \{\pi'\}$ and go to line 3 with $\pi \leftarrow \pi'$ {Start over with $\pi'$}
21:               **end if**
22:            **end if**
23:            $\mu(f) \leftarrow \succ_{f,j}$ and $\forall c \in A, \mu(c) \leftarrow d_0$ {$f$ is matched to $\succ_{f,j}$ and children $A$ are unmatched}
24:            $B \leftarrow \{c \in C^O \mid \mu^i(c) \neq \mu(c)\}$ {Children in $C^O$ matched differently under $\mu^i$ and $\mu$}
25:            $\mu^T \leftarrow \mu$ {Check whether $\mu$ is changed later}
26:            **while** $|B| > 0$ **do** {Stabilize children $B$}
27:               Choose one child $b \in B$ and initialize $h \leftarrow 1$
28:               **while** $h \leq |\succ_{f(b)}|$ **do**
29:                  $x \leftarrow \succ_{f(b),h}$, the $h$-th most preferred daycare in $\succ_{f(b)}$
30:                  $R \leftarrow \mu(x) \setminus \mathrm{Ch}_x(\mu(x) \cup \{b\})$
31:                  **if** $C(f) \cap R \neq \emptyset$ **then**
32:                      **return** Failure (Type-1)
33:                  **else if** $\exists f' \in F^S \setminus \{f\}$ s.t. $C(f') \cap R \neq \emptyset$ **then**
34:                      Go to line 16
35:                  **end if**
36:                  **if** $R = \{b\}$ **then**
37:                      $h \leftarrow h + 1$
38:                  **else**
39:                      $\forall c' \in R, \mu(c') \leftarrow d_0$ and $B \leftarrow B \cup \{c'\}$
40:                      $\mu(b) \leftarrow x$, $B \leftarrow B \setminus \{b\}$ and go to line 26
41:                  **end if**
42:               **end while**
43:               $B \leftarrow B \setminus \{b\}$
44:            **end while**
45:            **if** $\mu^T \neq \mu$ **then**
46:               Go to line 6 with $i \leftarrow i$ {Check $f$ one more time}
47:            **else**
48:               Update $\mu^{i+1} \leftarrow \mu$ and go to line 6 with $i \leftarrow i + 1$ {Check the next family in $F^S$}
49:            **end if**
50:         **end if**
51:      **end while**
52: **end while**
53: **return** A matching $\mu$.

**[Line 2-6]** Subsequently, the algorithm advances through $F^S$ based on the given order $\pi$. Consider the insertion of family $f = F^S_{\pi(i)}$ into the market, commencing with $i \leftarrow 1$. The following argument applies for any $i$ under the condition that no child from family $F^S_{\pi(i)}$ transfers seats to other siblings.

**[Line 7-12]** Family $f$ first applies to the tuple of daycares $\succ_{f,j}$, initialized with $j \leftarrow 1$ (line 7-8). If family $f$ cannot be accepted by all $d \in D(f, j)$, then the set of matched children at each daycare $d$ remains unchanged, i.e., $L(\mu, d)$ remains the same, and the algorithm proceeds to $j + 1$ (line 9-12).

**[Line 13]** If $D(f, j)$ still have vacant seats to accommodate family $f$, then we can imagine that dummy children are substituted by $C(f)$, resulting in a decrease in $L(\mu, d)$ at each daycare $d \in D(f, j)$. Subsequently, the algorithm proceeds to the next family $F^S_{\pi(i+1)}$.

**[Line 14]** Now, assume that some child is involved in the rejection chain $A$ during the insertion of family $f$. In this scenario, two possibilities arise.

**[Line 15-22]** Case i) If a child from another family $f' \in F^S \setminus \{f\}$ is rejected, it can lead to either a restart with a new permutation or result in a Type-2 Failure. In either case, it is equivalent to filling all seats at each daycare with dummy children assigned the rank $|C| + 1$, resulting in an increase in $L(\cdot)$. This indicates that the current order $\pi$ is unable to generate a matching.

**[Line 23-25]** Case ii) If only children in $C^O$ are affected during the insertion of $f$, we match $f$ to $\succ_{f,j}$ and assign any child in $A$ to the dummy daycare. In this scenario, $L(\cdot)$ decreases at each daycare $d \in D(f, j)$.

Let $B$ denote the set of children in $C^O$ matched differently under $\mu^i$ and $\mu$. We define $\mu^T$ as the matching before stabilizing the children in set $B$.

**[Line 26-35]** While stabilizing $B$, if a child from family $f'' \in F^S$ is rejected, the algorithm may either restart with a new permutation or terminate with failure. In either case, the current $\pi$ is inadequate for producing a matching, as discussed in Case i).

**[Line 36-44]** Next, let's consider the scenario where only children from $C^O$ are involved in $B$ during the stabilization process. In this case, if a child is rejected, it is replaced by another child with a higher priority, resulting in a decrease in $L(\cdot)$ at the corresponding daycare.

**[Line 45-49]** We need to verify whether $\mu$ differs from $\mu^T$ after stabilization. If they remain the same, $L(\cdot)$ does not change, and we proceed to the next family.

**[Back to Line 6-22]** Conversely, if $\mu$ differs from $\mu^T$, a supplementary check is conducted for family $f$ by allowing it to propose to $\succ_{f,j}$, staring with $j \leftarrow 1$. If family $f$ cannot be matched to a better tuple than $\mu^T(f)$, then $\mu$ as well as $L(\cdot)$ remain unchanged, and we move on to the next tuple.

Suppose family $f$ is matched to $\succ_{f,j}$ in matching $\mu^T$, and now family $f$ is matched to a better tuple denoted as $\succ_{f,k}$ in $\mu$. It's important to note that this scenario is possible because family $f$ is already matched under $\mu^T$, and some child can pass their seat to other siblings when reapplying to a better tuple than $\mu^T(f)$.

Formally, when family $f$ was rejected by $\succ_{f,k}$ in $\mu^T$, there must exist a daycare $d \in D(f, k)$, children $c, c' \in C(f)$, and a child $c^1 \in C^O$ such that: i) Child $c^1$, with the lowest priority, is matched to $d$ in $\mu^i$ (before processing family $f$). ii) The priority ordering at daycare $d$ satisfies: $c' \succ_d c^1 \succ_d c$. iii) Child $c'$ is matched to $\succ_{f,j}$ in $\mu^T$ by replacing $c^1$. When family $f$ reapplies to $\succ_{f,k}$ in matching $\mu$, child $c$ passes their seat to $c'$, resulting in an increase in $L(\mu, d)$.

**[Line 23-44]** Since child $c^1$ is matched differently under $\mu^i$ and $\mu$, we have $c^1 \in B$. When stabilizing $B$ again, child $c^1$ applies from their most preferred daycare. If $c^1$ reapplies to $d$, then it causes the rejection of $c$ and leads to a Type-1 Failure.

Let's assume that $c^1$ is matched to some daycare, say $d^1$, in $\mu$ which is more preferred than $d$, leading to an increase in $L(\mu, d^1)$. It's important to recall that $d^1$ was full under $\mu^i$ (before processing family $f$), and $d^1$ can accommodate $c^1$ in $\mu$ only if family $f$ causes some child $c^2$, who was matched to $d^1$ in $\mu^i$, to be affected in the rejection chain. Following the same argument, suppose $c^2$ could be matched to some daycare, say $d^2$, which is better than $d^1$, and $d^2$ was full under $\mu^i$ and some child $c^3$ was rejected when inserting $f$ under $\mu$.

Following the same argument, we can continue this chain until we reach a child, say $c^t$, who cannot be matched to a better daycare $d^t$ than $\mu^i(c^t)$ in $\mu$. If daycare $d^t$ has a vacant seat under $\mu$, it implies that $d^t$ must have had a vacant seat under $\mu^i$ before processing family $f$. However, this contradicts the fact that $c^t$ was rejected by $d^t$ under $\mu^i$. Therefore, all the children $c^t$, $c^{t-1}$, $c^{t-2}$, ..., $c^1$ could form a rejection chain ending with child $c$, leading to a Type-1 Failure.

Continuing this reasoning, we must arrive at some child, say $c^t$, who cannot be matched to a better daycare $d^t$ than $\mu^i(c^t)$ in this way. This is because family $f$ cannot create more vacancies than the number of children rejected by it when changing from $\succ_{f,k}$ to $\succ_{f,j}$, unless other families from $f' \in F^S \setminus \{f\}$ is rejected. However, in that case we will go to lines 15-22 instead. Therefore, we can conclude that the children $c^t$, $c^{t-1}$, $c^{t-2}$, $\cdots$, $c^1$, $c$ could form a rejection chain ending with child $c$, resulting in a Type-1 Failure.

Having meticulously examined all conceivable scenarios during the ESDA procedure, it is evident that $\pi$ is incapable of leading to a matching if $L(\mu, d)$ experiences an increase for any daycare $d$. This completes the proof of Lemma 7. $\square$

## D.2  Proof of Theorem 3

**Theorem 3.** *Given an instance of $I$, if ESDA returns a matching without failure, then the yielded matching is stable. In addition, ESDA always terminates in a finite time, either returning a matching or a failure.*

*Proof.* Suppose the ESDA in Algorithm 1 returns a matching $\mu$ without encountering any failures. Let $\tilde{\pi}$ denote the finial order over families $F^S$ when ESDA terminates.

Let $w = |F^S|$ denote the number of families in $F^S$, and consider the last family $f^w = F^S_{\tilde{\pi}(w)}$ in the order $\tilde{\pi}$. Case i) If family $f^w$ is matched to $\mu(f) = \succ_{f,j}$ without causing any child to be rejected, i.e., the stabilization step is not invoked, then for any $k \leq j$, family $f$ cannot be matched to a better tuple of daycares $\succ_{f,k}$, as the set of matched children remains unchanged at any $d \in D(f,k)$. Case ii) Suppose some children $A$ are rejected when inserting family $f^w$. We know $A \setminus F^S = \emptyset$, otherwise ESDA would terminate with a failure or restart with a new permuation. Thus $A \subseteq F^O$. After stabilizing all children $B$ (containing $A$) who are matched differently under $\mu^w$ and $\mu$, family $f$ reapplies to a better tuple of daycares by allowing for children $C(f)$ to pass their seats to other siblings. If this happens, then the rank of matched children $L(\cdot)$ at some daycare decreases, contradicting Lemma 7, which implies that $\tilde{\pi}$ can produce a matching. Thus, we know $f$ cannot be matched to a better tuple even if passing seats are allowed. For both cases, we conclude that family $f^w$ cannot pariticipate in a blocking coalition w.r.t. matching $\mu$.

Moving on to the second last family $f^{w-1}$, we apply a similar reasoning. When inserting family $f^{w-1}$ into the market, if it can be matched to a better tuple after the stabilization step, it contradicts Lemma 7. After family $f^w$ is introduced into the market, two key observations hold: i) the number of matched children does not decrease at any daycare, as per Lemma 6, and ii) for each daycare $d$, $L(\mu, d)$ does not increase, meaning no daycare accepts a child with a lower priority, per Lemma 7. Consequently, family $f^{w-1}$ still cannot be matched to a better tuple of daycares after the insertion of the last family $f$.

Continuing this logic through induction, we conclude that no family $f^i \in F^S_{\pi(i)}$ can be matched to a better tuple of daycares under the order $\tilde{\pi}$. In other words, none of the families in $F^S$ can participate in a blocking coalition. For the same reasons, it follows that any family $f \in F^O$ cannot be matched to a better daycare either.

For each permutation of $\pi$, the algorithm may iterate multiple times of checking $f$ for lines 45-46, if the current matching $\mu$ changes after the stabilization step. Since the choices in each only child's preference ordering are finite, the check terminates in a finite time or returns with a failure. Furthermore, the total number of permutations of $\pi$ is also finite, thus ensuring the algorithm's termination. This concludes the proof of Theorem 3. $\square$

**D.3  Two Types of Failure of ESDA**

**Example 7** (Type-1-a Failure). *Consider three families $f_1$ with children $C(f_1) = \{c_1, c_2\}$, $f_2$ with children $C(f_2) = \{c_3\}$ and $f_3$ with children $C(f_3) = \{c_4\}$. There are three daycares denoted as $D = \{d_1, d_2, d_3\}$, each with one available slot. The preferences of the families and the priorities of the daycares are outlined as follows:*

$$\succ_{f_1}: (d_1, d_3) \quad \succ_{f_2}: d_1, d_2 \quad \succ_{f_3}: d_2, d_1$$
$$\succ_{d_1}: c_4, c_1, c_3 \quad \succ_{d_2}: c_3, c_4 \quad \succ_{d_3}: c_2$$

*The initial matching $\mu^O$ is obtained through the Deferred Acceptance (DA) algorithm, where $\mu^O(c_3) = d_1$ and $\mu^O(c_4) = d_2$. Upon inserting family $f_1$, child $c_1$ is matched to daycare $d_1$, and child $c_2$ is matched to daycare $d_2$, resulting in the rejection of child $c_3$ from daycare $d_1$. Subsequently, when child $c_3$ applies to daycare $d_2$, it leads to the rejection of child $c_4$. Finally, when child $c_4$ applies to daycare $d_1$, it results in the rejection of child $c_1$.*

*Thus, a rejection chain is formed: $c_1 \to c_3 \to c_4 \to c_1$, and the ESDA algorithm terminates with failure. However, it's important to note that a stable matching $\mu'$ does exist, where $\mu'(c_3) = d_2$ and $\mu'(c_4) = d_1$. Despite of its existence, the ESDA algorithm fails to discover it.*

**Example 8** (Type-1-b Failure). *Consider two families $f_1$ with children $C(f_1) = \{c_1, c_2\}$ and $f_2$ with children $C(f_2) = \{c_3\}$. There are two daycares $D = \{d_1, d_2\}$, each having one available slot. The preferences of the families and the priorities of the daycares are outlined as follows:*

$$\succ_{f_1}: (d_1, d_2) \quad \succ_{f_2}: d_1, d_2$$
$$\succ_{d_1}: c_1, c_3 \quad \succ_{d_2}: c_3, c_2$$

*The initial matching $\mu^O$ is obtained through the Deferred Acceptance (DA) algorithm, with $\mu^O(c_3) = d_1$. Upon the introduction of family $f_1$, child $c_1$ secures a place at daycare $d_1$, and child $c_2$ is matched with daycare $d_2$, consequently leading to the rejection of child $c_3$ from daycare $d_1$. As child $c_3$ applies to daycare $d_2$, it results in the rejection of child $c_2$ from daycare $d_2$ in turn.*

*This sequence forms a rejection chain: $c_1 \to c_3 \to c_2$, prompting the ESDA algorithm to terminate with a failure. Notably, no stable matching is found to exist for Example 8.*

**Example 9** (Type-2 Failure). *Consider two families $f_1$ with children $C(f_1) = \{c_1, c_2\}$, and $f_2$ with children $C(f_2) = \{c_3, c_4\}$. There are three daycares, denoted as $D = \{d_1, d_2, d_3\}$, each with one slot. Suppose the initial order is $\pi = \{1, 2\}$. The preferences of the families and the priorities of the daycares are outlined as follows:*

$$\succ_{f_1}: (d_1, d_2), (d_1, d_3) \quad \succ_{f_2}: (d_2, d_3)$$
$$\succ_{d_1}: c_1 \quad \succ_{d_2}: c_3, c_2 \quad \succ_{d_3}: c_2, c_4$$

*When family $f_1$ is inserted, it secures a match with $(d_1, d_2)$. Subsequently, when family $f_2$ is added, child $c_2$ from family $f_1$ is rejected, prompting a change in the order to $\pi' = \{2, 1\}$ and a restart of the algorithm.*

*Now, if we add family $f_2$ first in the revised order $\pi'$, it obtains a match with $(d_2, d_3)$. However, when family $f_1$ is added and applies to $(d_1, d_2)$, child $c_2$ has a lower priority than child $c_3$, resulting in the rejection of family $f_1$. Consequently, family $f_1$ applies to $(d_1, d_3)$, causing family $f_2$ to be evicted in turn.*

*This leads us to modify the order $\pi'$ to $\pi^* = \{1, 2\}$, which has been attempted previously. Thus, the ESDA algorithm terminates due to a Type-2 Failure.*

# E   Proof of Theorem 1

In this section, we outline the proof for Theorem 1. Our main approach is to set an upper limit on the likelihood of encountering the two types of failure in the ESDA algorithm.

**Theorem 1.** *Given a random market $\tilde{I}$ with $\phi = O(\log n / n)$, the probability of the existence of a stable matching converges to $1$ as $n$ approaches infinity.*

We leverage the following lemma in our proof. It asserts that if an ordering $\succ$ is generated from a given Mallows distribution $\mathcal{D}_{\succ_0, \phi}$, the probability of child $c'$ being ranked higher than child $c$ in $\succ$ is no greater than $4\phi^{\text{dist}(c,c')}$, given that $c \succ_0 c'$, where $\text{dist}(c, c')$ represents the distance between $c$ and $c'$ in $\succ_0$.

**Lemma 8** ([Levy, 2017]). *If $\succ$ is a random ordering drawn from the Mallows distribution $\mathcal{D}_{\succ_0, \phi}$, then for all $c, c' \in C$,*

$$\Pr\big[c' \succ c \mid c \succ_0 c'\big] \leq 4\phi^{\text{dist}(c,c')}$$

*where $\text{dist}(c, c') = |\{c'' \in C \mid c \succ_0 c'' \succ_0 c'\}| + 1$.*

### E.1 Proof of Lemma 1

**Lemma 1.** *Under the uniformly bounded condition, the probability $p_d$ of selecting any daycare $d$ is limited by $\sigma/m$ where $m$ denotes the total number of daycares.*

*Proof.* For each daycare $d$, we have $1/\sigma \leq p_d/p_{d'} \leq \sigma$. Therefore, $p_{d'}/\sigma \leq p_d \leq \sigma \cdot p_{d'}$. If we sum this inequality over each $d' \in D$, we obtain $m \cdot p_d \leq \sum_{d' \in D} \sigma \cdot p_{d'} = \sigma$. Thus, $p_d \leq \sigma/m$. □

### E.2 Proof of Lemma 2

**Lemma 2.** *Given a random market $\tilde{I}$ with $\phi = O(\log n/n)$, the probability of Type-$1$-a Failure in the SDA algorithm is bounded by $O\big((\log n)^2/n\big)$.*

*Proof.* We first consider a Type-1-a Failure, where a rejection chain $c_1 \rightarrow c_2^* \rightarrow \cdots \rightarrow c_\ell^* \rightarrow c_1$ exists. Here, child $c_1$ belongs to a family $f \in F^S$ with multiple children, while the other children $c_2^*, \cdots, c_\ell^* \in C^O$ have no siblings.

Let $\mathcal{E}_\ell^{\text{a}}$ represent the event of such a rejection chain $c_1 \rightarrow c_2^* \rightarrow \cdots \rightarrow c_\ell^* \rightarrow c_1$, with length $\ell \geq 3$. We next show that, for any $\succ_0$, we have

$$\Pr[\mathcal{E}_\ell^{\text{a}} \mid \succ_0] \leq \frac{16\sigma\phi^2}{m}. \tag{1}$$

Suppose that in this rejection chain, child $c_1$ applies to daycare $d_1$, while children $c_i^*$ apply to $d_i^*$ for $i \in \{2, 3, ..., \ell - 1\}$. The last child in the cycle, $c_\ell^*$, applies to daycare $d_1$. It is important to note that $d_i^* \neq d_{i+1}^*$ holds for $i \in \{1, \ldots, \ell - 2\}$, even though there could be repetitions among the children $c_2^*, ..., c_\ell^*$ and the daycares $d_2^*, ..., d_{\ell-1}^*$.

Let $\succ_1$ represent the priority ordering of daycare $d_1$. For $i \in \{2, \ldots, \ell - 1\}$, let $\succ_i$ denote the priority ordering of daycare $d_i^*$. Recall that for each $i = 1, \ldots, \ell - 1$, the priority ordering $\succ_i$ is drawn from the Mallows distribution $\mathcal{D}_{\succ_0, \phi}$. We consider two cases.

Case (i): Suppose the reference ordering $\succ_0$ satisfies the following condition

$$c_\ell^* \succ_0 c_{\ell-1}^* \succ_0 \cdots \succ_0 c_2^* \succ_0 c_1. \tag{2}$$

By Lemma 8, we have

$$\Pr[c_\ell^* \succ_1 c_1 \succ_1 c_2^* \mid \succ_0] \leq \Pr[c_1 \succ_1 c_2^* \mid c_2^* \succ_0 c_1] \leq 4\phi.$$

For all $i = 2, ..., \ell - 1$, we also have

$$\Pr[c_i^* \succ_i c_{i+1}^* \mid \succ_0] \leq 4\phi.$$

From $d_1^* \neq d_2^*$, we know $\succ_1$ and $\succ_2$ are independent. Then we have

$$\Pr\big[\mathcal{E}_\ell^{\text{a}} \mid \succ_0\big] \leq \Pr\big[c_1 \succ_1 c_2^* \mid \succ_0\big] \cdot \Pr\big[c_2^* \succ_2 c_3^* \mid \succ_0\big] \cdot \Pr\big[c_{\ell-1}^* \text{ applies to } d_1\big]$$
$$\leq 16\phi^2 p_{d_1}.$$

Lemma 1 states that $p_{d_1} \leq \sigma/m$. Then we have

$$\Pr\big[\mathcal{E}_\ell^{\text{a}} \mid \succ_0\big] \leq 16\phi^2 p_{d_1} \leq \frac{16\sigma\phi^2}{m}. \tag{3}$$

Case (ii): If $\succ_0$ does not satisfy the condition in Formula (2), then $\Pr[c_\ell^* \succ_1 c_1 \succ_1 c_2^* \,|\succ_0] \le 4\phi^2$ holds or there exists $i \in \{2, ..., \ell - 1\}$ such that $\Pr[c_i^* \succ_i c_{i+1}^* \,|\succ_0] \le 4\phi^2$. From this, we obtain

$$\Pr\big[\mathcal{E}_\ell^{\mathrm{a}} \,|\succ_0\big] \le 4\phi^2 \cdot \Pr\big[c_{\ell-1}^* \text{ applies to } d_1\big]$$
$$\le 4\phi^2 p_{d_1}$$
$$\le \frac{4\sigma\phi^2}{m}. \tag{4}$$

From Inequalities (3) and (4) above, for both cases (i) and (ii), we have $\Pr[\mathcal{E}_\ell^{\mathrm{a}} \,|\succ_0] \le \frac{16\sigma\phi^2}{m}$. This completes the proof of Inequality (1).

Given that $\succ_0$ is drawn from a uniform distribution over all permutations of $C$, we can derive the following inequality for the probability of encountering Type-1-a Failure, denoted as $\mathcal{E}_\ell$, for a particular length $\ell$ of the rejection chain:

$$\Pr\big[\mathcal{E}_\ell^{\mathrm{a}}\big] \le \sum_{\succ_0 \in S'} \Pr\big[\mathcal{E}_\ell^{\mathrm{a}} \,|\succ_0\big] \cdot \Pr[\succ_0]$$
$$\le \frac{16\sigma\phi^2}{m} \sum_{\succ_0 \in S'} \Pr[\succ_0]$$
$$= \frac{16\sigma\phi^2}{m}$$

where $S'$ denotes all permutations on the set of children $C$ that is used to generate $\succ_0$.

To obtain the overall probability of Type-1-a Failure, we sum up the probabilities for all possible lengths $\ell$ and for all children $F^S$. Recall that the length of each child's preference ordering is bounded by $L$, and the length of a rejection chain is upper bounded by $(1-\alpha)n \cdot L$ and lower bounded by 3. Thus, the probability that there exists a rejection cycle leading Type-1-a Failure is bounded from above by

$$\alpha n \cdot \sum_{\ell=3}^{(1-\alpha)nL} \Pr\big[\mathcal{E}_\ell^{\mathrm{a}}\big] \le 16\alpha(1-\alpha)L\sigma \frac{n^2\phi^2}{m}.$$

If $\phi = O(\log n/n)$, the probability of there being a Type-1-a Failure is $O\left(\frac{(\log n)^2}{n}\right)$, which converges to 0 as $n$ approaches infinity. $\qquad\square$

### E.3 Proof of Lemma 3

**Lemma 3.** *Given a random market $\tilde{I}$ with $\phi = O(\log n/n)$, the probability of Type-1-b Failure in the SDA algorithm is bounded by $O\big((\log n)^2/n\big) + O(n^{-\varepsilon})$.*

*Proof.* We next proceed to Type-1-b Failure, where a rejection chain is denoted as $c_1 \to c_2^* \to \cdots \to c_\ell^* \to c_1'$. Here, $c_1$ and $c_1'$ are siblings of the same family $f \in F^S$, while $c_2^*, \ldots, c_\ell^*$ are children without siblings. Suppose that $c_i^*$ applies to $d_i^*$ for each $i = 2, 3, ..., \ell - 1$.

If children $c_1$ and $c_1'$ have nearly identical priorities in $\succ_0$ ($\mathrm{diam}_f \le |C(f)|$), the analysis aligns with that of Type-1-a Failure. Consequently, in this scenario, the probability of the rejection chain occurring is at most $16\sigma\phi^2/m$ for any $\succ_0$ and for any $2 \le \ell \le (1-\alpha)nL$.

If children $c_1$ and $c_1'$ have significantly different priorities in $\succ_0$ ($\mathrm{diam}f > |C(f)|$), then it only occurs with a probability at most $1/n^{1+\varepsilon}$ ($\varepsilon > 0$). Therefore, even in the worst-case scenario where $\succ_0$ satisfies $c_1^* \succ_0 c_2^* \succ_0 \cdots \succ_0 c_\ell^* \succ_0 c_1'^*$, the probability that the last child $c_\ell^*$ causes $c_1'$ to be rejected, is bounded by $\frac{\sigma}{n^{1+\varepsilon}m}$.

Let $\mathcal{E}_\ell^{\mathrm{b}}$ denote the event where the rejection chain of length $\ell$ starting with $c_1$ and ending with $c_1'$ occurs. For any $\ell$ and $\succ_0$, we have

$$\Pr\big[\mathcal{E}_\ell^{\mathrm{b}} \,|\succ_0\big] \le \frac{16\sigma\phi^2}{m} + \frac{\sigma}{n^{1+\varepsilon}m}.$$

We next sum up the probabilities for all possible lengths $\ell$ and for any two children in families with multiple children. The probability of Type-1-b Failure occurring is bounded by

$$\alpha n \cdot \binom{\bar{k}}{2} \cdot \sum_{\ell=2}^{(1-\alpha)nL} \Pr\left[\mathcal{E}_\ell^b\right]$$

$$\leq \alpha(1-\alpha)L\bar{k}^2 n^2 \left(\frac{16\sigma\phi^2}{m} + \frac{\sigma}{n^{1+\varepsilon}m}\right)$$

$$= O\left(\frac{(\log n)^2}{n}\right) + O(n^{-\varepsilon}).$$

Here, we used $m = \Omega(n)$ and $\phi = O(\log n/n)$. This concludes that Type-1 Failure does not happen with high probability. $\qquad\square$

## E.4 Proof of Lemma 4

In addition to the concept of domination, we define the notion of *top-domination*.

**Definition 14** (Top Domination). *Given a priority ordering $\succ$, we say that family $f$ top-dominates $f'$ w.r.t. $\succ$ if*

$$\max_{c\in C(f)} c \succ \max_{c'\in C(f')} c'.$$

**Lemma 4.** *Given a random market $\tilde{I}$ with $\phi = O(\log n/n)$, and for any two families $f, f' \in F^S$ that are not nesting with each other with respect to $\succ_0$, then Type-2 Failure occurs with a probability of at most $O(\log n/n)$.*

*Proof.* Consider any two families $f, f' \in F^S$ that do not nest with each other. Without loss of generality, we assume that $f$ top-dominates $f'$, and $f'$ does not dominate $f$, otherwise they would nest with each other. Then we have,

$$\forall c \in C(f), \forall c' \in C(f'), c \succ_0 c'. \tag{5}$$

Suppose $f'$ appears before $f$ in the order $\pi$ over families $F^S$, and $f'$ is currently matched. When $f$ is inserted into the market, we observe that the probability of $f$ causing the rejection of $f'$ is bounded by $\sigma/m$, i.e., $\Pr\left[f \text{ rejects } f'\right] \leq \sigma/m$, given that preferences are uniformly bounded.

Next, consider a new order $\pi'$ in which $f$ is placed before $f'$. We aim to analyze the probability of $f'$ causing the rejection of $f$ in a rejection chain of length $\ell$.

We begin with $\ell = 2$. Suppose a child $c \in C(f)$ is currently matched to daycare $d_1$, and another child $c' \in C(f')$ also applies to daycare $d_1$, resulting in the rejection of child $c$. As shown in Formula (5), we have $c \succ_0 c'$. Since $c' \succ_1 c$, we can deduce that $\Pr[c' \succ_1 c \,|\succ_0] \leq 4\phi$ from Lemma 8.

Let $\mathcal{E}'_0$ be the event where $f$ rejects $f'$, followed by $f'$ rejecting $f$. The probability that one child in $C(f')$ applies to $d_1$ is upper-bounded by $\sigma/m$. Therefore, we can derive:

$$\Pr\left[\mathcal{E}'_0\right] \leq \left(\frac{\sigma}{m}\right)^2 4\phi = \frac{4\sigma^2\phi}{m^2}.$$

Next, we consider the scenario where a rejection chain of length $\ell + 2$ occurs, where $\ell$ represents the number of children without siblings participating in the rejection chain. Suppose the rejection chain follows the pattern $c \to c_1^* \to c_2^* \to \cdots \to c_\ell^* \to c'$, where $c_1^*, ..., c_\ell^* \in C^O$. In this case, we have $1 \leq \ell \leq (1-\alpha)nL$.

Let $\mathcal{E}'_\ell$ be the event where $f$ rejects $f'$, and subsequently $f'$ rejects $f$ using a rejection chain of length $\ell$. For any $\succ_0$, the replacement by the Mallows distribution must happen at least twice. Thus, for each $\ell = 1, 2, \ldots, (1-\alpha)nL$, we have

$$\Pr\left[\mathcal{E}'_\ell \,|\succ_0\right] \leq \left(\frac{\sigma'}{m}\right)^2 16\phi^2 \leq \frac{16\sigma'\phi^2}{m^2}.$$

We sum up the probabilities for all possible $\succ_0$, and achieve $\Pr\left[\mathcal{E}'_\ell\right] \leq \frac{16\sigma'\phi^2}{m^2}$ for each $\ell = 1, 2, \ldots, (1-\alpha)nL$. Then we obtain

$$\sum_{\ell=1}^{(1-\alpha)nL} \Pr\left[\mathcal{E}'_\ell\right] \leq \frac{16(1-\alpha)L\sigma n\phi^2}{m^2}.$$

Finally, since $m = \Omega(n)$ and $\phi = O(\log n/n)$, we get

$$\Pr\left[\text{there exists a pair of families with siblings cause rejections with each other}\right]$$

$$= \sum_{f,f'\in F^S} \Pr\left[\bigcup_{\ell=0}^{(1-\alpha)\bar{k}n} \mathcal{E}'_\ell\right]$$

$$\leq \sum_{f,f'\in F^S} \sum_{\ell=0}^{(1-\alpha)nL} \Pr\left[\mathcal{E}'_\ell\right]$$

$$= \sum_{f,f'\in F^S} \left(\Pr\left[\mathcal{E}'_0\right] + \sum_{\ell=1}^{(1-\alpha)nL} \Pr\left[\mathcal{E}'_\ell\right]\right)$$

$$\leq (\alpha n)^2 \left(\frac{16\sigma\phi}{m^2} + \frac{16(1-\alpha)\bar{k}\sigma n\phi^2}{m^2}\right)$$

$$= O\left(\frac{\log n}{n}\right). \qquad \square$$

## E.5    Proof of Lemma 5

**Lemma 5.** *Given a random market $\tilde{I}$ with $\phi = O(\log n/n)$, the probability of Type-2 Failure occurring is bounded by $O(\log n/n) + O\left(n^{-2\varepsilon}\right)$.*

*Proof.* We first consider the probability that any two pairs of families with multiple siblings nest with each other w.r.t. the reference ordering $\succ_0$.

For any two families $f$ and $f'$, if they nest with each other, then the diameters of both $f$ and $f'$ are large, i.e., $\text{diam}_f > |C(f)|$ and $\text{diam}_{f'} > |C(f')|$. Thus, the inequality $\Pr\left[\text{diam}_f \geq |C(f)|\right] \leq \frac{1}{n^{1+\varepsilon}}$ implies that

$$\Pr\left[f \text{ and } f' \text{ nest with each other}\right] \leq \left(\frac{1}{n^{1+\varepsilon}}\right)^2.$$

Hence, we have

$$\Pr\left[\text{there exist two families who nest with each other}\right]$$

$$\leq \sum_{f,f'\in F^S} \Pr\left[f \text{ and } f' \text{ nest with each other}\right]$$

$$\leq \binom{\alpha n}{2} \cdot \left(\frac{1}{n^{1+\varepsilon}}\right)^2$$

$$\leq \alpha^2 n^2 \cdot \left(\frac{1}{n^{1+\varepsilon}}\right)^2$$

$$= O\left(n^{-2\varepsilon}\right).$$

Since $\varepsilon > 0$ is a constant, the probability that any two families do not nest with each other approaches 1 as $n$ tends to infinity.

We now upper-bound the probability of Type-2 Failure. In cases where two families nest with each other, Type-2 Failure may occur with a constant probability. However, we have demonstrated that the probability of two families nesting with each other is at most $O(n^{-2\varepsilon})$. In instances where no two

families nest with each other, Type-2 Failure happens with a probability of at most $O(\log n/n)$ as shown in Lemma 4. Therefore, we can express the probability of Type-2 Failure as follows:

$$\Pr\big[\text{Type-2 Failure happens}\big] = O\big(n^{-2\varepsilon}\big) + O(\log n/n).$$

This completes the proof. $\qquad\square$

Lemma 2, 3 and 5 imply the existence of a stable matching with high probability for the large random market, thus concluding the proof of Theorem 1.

## F  More on Experiments

### F.1  Features of Real-life Markets

We are collaborating with several municipalities in Japan, and as part of our collaboration, we provide a detailed description of the practical daycare matching markets based on data sets provided by three representative municipalities.

Firstly, the number of children in each market varies from $500$ to $1600$, with the proportion of children having siblings consistently spanning from $15\%$ to $20\%$, as shown in Table 1.

|  | fraction | # children |
|---|---|---|
| Shibuya 21 | 16.24% | 1589 |
| Shibuya 22 | 15.38% | 1372 |
| Tama 21 | 16.45% | 635 |
| Tama 22 | 16% | 550 |
| Koriyama 22 | 20.68% | 1383 |
| Koriyama 23 | 19.14% | 1458 |

Table 1: Fraction of children with siblings. This table presents the proportion of children with siblings, along with the total number of children in each dataset.

Secondly, the preference ordering of an only child is relatively short compared to the available facilities, averaging between 3 and 4.5 choices. Likewise, children from families with siblings exhibit a similar average of 3 to 4.5 distinct daycares in their individual preferences. Furthermore, siblings within the same family often share a similar set of daycares in their joint preference ordering. The details are presented in Table 2.

| length | only | sibling | distinct |
|---|---|---|---|
| Shibuya 21 | 4.45 | 14.86 | 4.26 |
| Shibuya 22 | 3.76 | 6.58 | 3.64 |
| Tama 21 | 3.29 | 38.29 | 3.43 |
| Tama 22 | 3.01 | 8.55 | 3.17 |
| Koriyama 22 | 3.02 | 21.38 | 3.60 |
| Koriyama 23 | 3.10 | 9.42 | 3.13 |

Table 2: Average length of preferences. The second column pertains to families with only one child, while the third column represents families with siblings. The last column displays the average number of distinct daycares in the corresponding individual preference lists for children with siblings.

Thirdly, a critical aspect not mentioned in Section 3.1 is that each child is associated with an age ranging from $0$ to $5$. Drawing inspiration from prior work [Sun et al., 2023], we make the assumption that there are six copies of the same daycare, each catering to a specific age. The distribution of children participating in the market is uneven, with a notable majority being aged $0$ and $1$. In Table 3, we present the count of families with siblings and twins (i.e., pairs of siblings of the same age).

Fourthly, despite the total capacity of all daycares exceeding the number of applicants, there exists a significant imbalance between demand and supply across different ages. Specifically, there is a shortage of slots for children aged $0$ and $1$, while there is a surplus of slots for ages $4$ and $5$, as shown in Table 4.

|  | # children in the family | | | |
|  | 2 | | $\geq 3$ | |
| # families | total | twin | total | twin |
| --- | --- | --- | --- | --- |
| Shibuya 21 | 120 | 14 | 6 | 4 |
| Shibuya 22 | 101 | 25 | 3 | 3 |
| Tama 21 | 42 | 3 | 3 | 3 |
| Tama 22 | 44 | 8 | 0 | 0 |
| Koriyama 22 | 123 | 10 | 13 | 2 |
| Koriyama 23 | 130 | 12 | 6 | 0 |

Table 3: Number of families with siblings and twins. The second and third columns represent families with 2 children, while the last two columns represent families with 3 or more children.

| | age | 0 | 1 | 2 | 3 | 4 | 5 |
| --- | --- | --- | --- | --- | --- | --- | --- |
| Shibuya-21 | # applicants | 569 | 656 | 171 | 136 | 37 | 20 |
| | # capacity | 509 | 613 | 239 | 265 | 268 | 275 |
| Shibuya-22 | # applicants | 540 | 582 | 134 | 67 | 33 | 16 |
| | # capacity | 497 | 586 | 186 | 233 | 255 | 306 |
| Tama-21 | # applicants | 181 | 257 | 98 | 75 | 17 | 7 |
| | # capacity | 241 | 222 | 123 | 106 | 57 | 68 |
| Tama-22 | # applicants | 181 | 219 | 91 | 43 | 8 | 8 |
| | # capacity | 231 | 218 | 100 | 97 | 45 | 47 |
| Koriyama-22 | # applicants | 379 | 538 | 140 | 231 | 59 | 36 |
| | # capacity | 546 | 585 | 220 | 327 | 276 | 171 |
| Koriyama-23 | # applicants | 366 | 588 | 167 | 239 | 64 | 33 |
| | # capacity | 559 | 511 | 218 | 282 | 139 | 188 |

Table 4: Demand and supply by age

Fifthly, municipalities assign priority scores to children, with siblings from the same family typically sharing identical scores. Subsequently, daycares make slight adjustments to these priority scores to establish a strict priority ordering. As a result, all daycares tend to have similar priority orderings over the children.

## F.2  More Experiments

We employ both the Extended Sorted Deferred Acceptance (ESDA) algorithm and the constraint programming (CP) algorithm to find a stable matching for each real-life dataset. The results demonstrate that both algorithms successfully produce a stable matching. We compared the computational efficiency of the ESDA and CP approaches in terms of their runtime performance in Table 5.

In the experiments with synthetic datasets, the ESDA algorithm consistently identifies a stable matching whenever one exists, provided that the dispersion parameter $\phi$ does not exceed $0.5$ (refer to Figure 2 in Section 7.2). However, as the dispersion parameter approaches $1$, the ESDA algorithm may fail to find a stable matching, even when one exists. This is illustrated in Figure 3. Interestingly, even when $\phi = 1$, stable matchings are present in more than half of the cases. It is unclear why stable matching still exist in such settings with a high probability, and we leave it as an open question.

Table 5: Results of computation times (seconds) for experiments on real-world data.

| | ESDA | CP |
| --- | --- | --- |
| Shibuya 21 | 0.87 | 13.08 |
| Shibuya 22 | 0.50 | 8.17 |
| Tama 21 | 0.10 | 7.33 |
| Tama 22 | 0.07 | 1.41 |
| Koriyama 22 | 0.50 | 14.10 |
| Koriyama 23 | 0.65 | 6.57 |

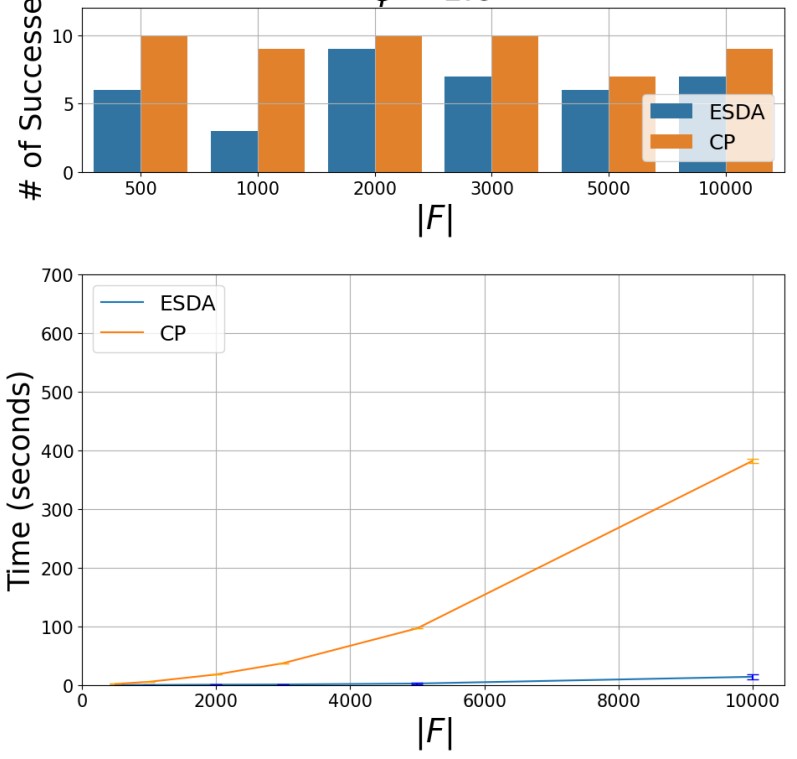

Figure 3: Results of experiments on synthetic data when $\phi = 1.0$.

