# OpenReview forum: "Probabilistic Analysis of Stable Matching in Large Markets with Siblings"
_NeurIPS.cc/2024/Conference — Submitted to NeurIPS 2024_

### Official Review · Reviewer_52vJ · 2024-06-21

**Soundness:** 3
**Presentation:** 2
**Contribution:** 3
**Rating:** 6
**Confidence:** 3

**Summary:**

The paper describes the problem of matching students to daycare centers, with each family allowed to express preferences about the joint allocation of all siblings within the family. The authors present a modified notion of a stable matching, in which a family may choose to withdraw one of its children from a daycare in favor of a different child, so long as the daycare still prefers this assignment among alternatives with the first child removed. Under this stronger notion of stability, they show that the existing SDA algorithm may produce unstable outputs. They present an extension to the algorithm ESDA, whose successful outputs meet the new stability condition. They also show that the algorithm will be successful with high probability for a particular distribution over problem instances. In this distribution, children populate a daycare preference order by selecting from a fixed distribution over the daycares, and families aggregate these preferences into preferences over joint allocations, using an arbitrary aggregation function. Daycares sample a preference order over children from a Mallows model, with low dispersion.
Finally, the authors show some empirical results from real Japanese municipalities in which ESDA produces stable matchings in all cases.

**Strengths:**

The algorithm is a heuristic (for good reasons of computational hardness of the problem). Hence, I characterize the contributions as follows:
* The authors define the problem, generalizing from couple-matching instances that may be expressed as families of size at most 2
* The authors present a new notion of stability in the matching which seems to be justified, given that families are generally empowered to prevent one of their children from attending a particular daycare.
* The authors present the ESDA algorithm, which internalizes the new stability notion in the details of the algorithm execution, and produces stable matchings whenever the algorithm succeeds
* Empirical analysis shows the algorithm succeeding on real-world instances
* Finally, the authors show that real-world instances have some strong properties in terms of the similarity across daycares of the preference ordering for students. They also incorporate this observation into the algorithm design, and are able to show that under a certain random model of problem instances, the algorithm succeeds with high probability. In the real-world examples, the preferences of the daycare are largely provided by the municipality, so the assumption is very likely to hold.
* As a smaller point, I appreciate that the authors presented some analysis of the behavior of the algorithm when the dispersion of the mallows models becomes high.
* An additional smaller point: earlier results that operate with a vanishing fraction of couples in the population seem unsatisfying. The theoretical results in this paper instead allow a constant fraction of the families to have siblings, but place stronger constraints on the similarity of orderings of the daycares, which seems better justified.

**Weaknesses:**

My first question is about goodness of fit of the paper to NeurIPS. The best fit from the CFP is:
* Social and economic aspects of machine learning (e.g., fairness, interpretability, human-AI interaction, privacy, safety, strategic behavior)
specifically for strategic behavior. However, I'm not sure this should be called economic aspects *of machine learning* specifically. I'll leave this issue with the area chair---my personal view is that it's not a great match. There are some related papers that have appeared in past conference instances (for instance, on deferred acceptance variants, but with more of a focus on computational complexity of an algorithmic approach, such as https://papers.nips.cc/paper_files/paper/2019/hash/cb70ab375662576bd1ac5aaf16b3fca4-Abstract.html).

Second, I'm concerned about the family preference model in section 4.1. In particular, the title of the paper says "Large Markets," and as the size of the market grows beyond a small geographic area, it seems like that geographic preferences (for nearby daycares) will play a role. However, the random model is based on a single global distribution of preferences that applies to all families across all locations. This distribution is then further constrained to place similar probabilities on all daycares. There is no analysis of the empirical data to justify this uniformity assumption. Additionally, the model assigns independent preferences to two siblings of the same family, which seems to miss a) the fact that a family may have certain specific desires, and b) the family's geo preferences will apply similarly to all children, and c) sending multiple siblings to the same daycare may provide complementarities such as less logistic overhead for transport.

**Questions:**

I'd love to hear responses from the authors on the two points I raise in the Weaknesses section above.

A few smaller points:

Line 113: the definition of a matching as mapping from C \cup D to C \cup D seems off -- is the intention to map to C\cup 2^{D}? Also, \mu(c) is a daycare (or a singleton set containing a daycare?), while \mu(d) is a set of children? Maybe I'm missing the structure, but this seems unclear.

Definition 1: what is the motivation for each family to be matched either to a tuple from their preference order or the unassigned tuple? Why is it not preferred to have an assignment for just some children in the family, versus none (d_0,\ldots,d_0)?

**Limitations:**

I think the authors have done a good job here.

---

> ### Author Rebuttal · Authors · 2024-08-03
>
> Thank you so much for your detailed comments and questions.
>
> Q1
> We fully understand your concern regarding the suitability of this paper for a top-tier ML conference. We have chosen the Theory area, specifically Algorithmic Game Theory, where stable matchings have demonstrated significant applications. One of our primary goals in submitting this paper to NeurIPS is to attract the attention and feedback of a broader AI community and to explore potential collaboration opportunities.
>
> Here are some NeurIPS papers that do not focus on computational complexity:
>
> https://proceedings.neurips.cc/paper/2020/hash/7e05d6f828574fbc975a896b25bb011e-Abstract.html
> https://proceedings.neurips.cc/paper_files/paper/2023/hash/ccba10dd4e80e7276054222bb95d467c-Abstract-Conference.html
> https://proceedings.neurips.cc/paper_files/paper/2022/hash/17bb0edcc02bd1f74e771e23b2aa1501-Abstract-Conference.html
>
> Q2
> In our theoretical analysis, we consider the general case where siblings' preferences are independent. We show that even in this scenario, the proposed algorithm can find a stable matching with high probability when the number of children is sufficiently large.
>
> As you suggested, incorporating practical assumptions into family preferences could provide a more precise bound in the probability analysis. This is indeed our next research question to explore.
>
> Line 113:  We will rewrite the definition of a matching as follows: $\mu : C \rightarrow D$.
>
> Definition 1: A family may only consider certain tuples of daycares acceptable. Sending some children rather than all children to some daycare is possible, with the least preferred option being to send none. For example, if a daycare is too far from their home, they may prefer not to send any of their children there.

---

> > ### Comment · Reviewer_52vJ · 2024-08-07
> > **Request for clarification**
> >
> > Thanks to the authors for their response. I didn't fully understand the response to the following question:
> >
> > ----
> > Second, I'm concerned about the family preference model in section 4.1. In particular, the title of the paper says "Large Markets," and as the size of the market grows beyond a small geographic area, it seems like that geographic preferences (for nearby daycares) will play a role. However, the random model is based on a single global distribution of preferences that applies to all families across all locations. This distribution is then further constrained to place similar probabilities on all daycares. There is no analysis of the empirical data to justify this uniformity assumption. Additionally, the model assigns independent preferences to two siblings of the same family, which seems to miss a) the fact that a family may have certain specific desires, and b) the family's geo preferences will apply similarly to all children, and c) sending multiple siblings to the same daycare may provide complementarities such as less logistic overhead for transport.
> > ----
> >
> > I'm particular wondering here about a scenario in which the algorithm is applied to a broad area, each family ranks daycares that are within walking distance, versus those that require an hour's drive across the city, and will in many cases prefer a closer one. I believe the random model assumes draws from a global preference distribution of daycares (whether independently drawn for siblings or not), where I would expect the reality to be much closer to a distribution that incorporates preferences for nearby daycares. I may well be misunderstanding something in the model or response -- clarification appreciated.

---

> > > ### Author Response · Authors · 2024-08-09
> > >
> > > Firstly, we do not have access to the geographical information of families, as it is private and confidential. Consequently, we were unable to generate a distribution that incorporates preferences for nearby daycares based on the data sets we have.
> > >
> > > Secondly, we followed the approach outlined by Kojima et al. (2013), where the joint preferences of siblings are derived from a random distribution seperately. In this paper, our aim is to understand how common priorities among daycares influence the existence of stable matchings.
> > >
> > > Lastly, we fully agree with your observation that the pattern in families’ preferences also impacts the existence of stable matchings. We will address this aspect with a more nuanced analysis in the next phase of our research.

---

> > > > ### Comment · Reviewer_52vJ · 2024-08-09
> > > > **Followup on geo**
> > > >
> > > > Understood about the issue that geo location of families is not available in the dataset (for privacy reasons). And to be clear on the high-order bit, I'm still in favor of accepting this paper. But I'd like to follow up on this a little.
> > > >
> > > > First, I think the authors' final point is a good one. It's not clear (at least to me) that distributed markets with strong geo preferences should be fundamentally easier or harder than markets without -- it's possible that problem instances with strong geo preferences behave like a number of smaller problem instances, one for each geographic region, which is clearly useful structure. Likewise for other forms of family preference.
> > > >
> > > > Next, let's touch on the theoretical discussion (section 4, and in particular, Theorem 1). I believe this is the only place that the assumption that a fixed distribution $\cal P$ is used to seed individual preferences for daycares. Theorem 1 is nice, and seems to me to be powerful compared to earlier results. My question here is more for the future than an ask for the current paper: is it possible to strengthen the theorem to include a notion of proximity. A standard approach, for example, would be to assume households and daycares are located uniformly in a unit square, and that the student preference distribution includes a convenient term based on the distance between the household and the daycare. There is perhaps a loose intuition that for sufficiently large n, the theorem should still hold, as the preferences for the (diverging) number of nearby households will look almost identical, again given $m=\Omega(n)$. This doesn't really need a response -- it's more of a passing observation.
> > > >
> > > > When it comes to the empirical data, here we see the algorithm working very well, and there are two possibilities:
> > > > 1. The municipalities, for some reason, have family preferences that are inline with Theorem 1, so do not have a strong geo effect or other heterogeneous family preferences. This is interesting because it shows that Theorem 1 aligns with real-world cases.
> > > > 2. Family preferences are in fact heterogeneous, perhaps for geographic reasons, but the algorithm continues to work well, beyond the bounds of Theorem 1. This is also interesting, because it shows the applicability of the algorithm in practice is broader than the theorem.
> > > >
> > > > Does this characterization sound correct? If so, do you have any sense for which case holds? It seems that the correlation of sibling preferences, compared to the correlation of non-sibling preferences should given an indication of case 1 versus 2?
> > > >
> > > > To summarize, I am supportive, and any additional understanding of the nature of the assumptions in section 4.1 (how much they are required by theorem 1, and how much they entail in practice) only increases my confidence in the generality of the results. Thank you for any additional thoughts on this subject.

---

> > > > > ### Author Response · Authors · 2024-08-12
> > > > > **Followup on geo**
> > > > >
> > > > > > My question here is more for the future than an ask for the current paper: is it possible to strengthen the theorem to include a notion of proximity.
> > > > >
> > > > > Thank you for your insight. If locations are selected uniformly at random in a unit square, it may result in preferences being uniformly randomly distributed. Thus, we believe that the distribution satisfies Definition 5 (Uniformly Bounded), and our results are applicable. A potential avenue for future research would be to consider a model where locations are deterministically given, and preferences are generated by adding randomness to deterministic preferences based on these fixed locations. In such a scenario, the assumption of a fixed distribution $\mathcal{P}$ (Definition 5: Uniformly Bounded) may not hold.
> > > > >
> > > > > > Does this characterization sound correct? If so, do you have any sense for which case holds? It seems that the correlation of sibling preferences, compared to the correlation of non-sibling preferences should given an indication of case 1 versus 2?
> > > > >
> > > > > Currently, we cannot determine the details. For Case 1 (Alignment with Theorem 1), individual municipalities are geographically large with childcare centers in various locations, so preferences might align well with the Uniformly Bounded. On the other hand, for Case 2 (Effectiveness beyond Theorem 1), the algorithm may work effectively beyond this assumption. For instance, in our experiments, when we slightly and adversarially altered family preferences from their current state, stable matchings continued to exist. However, when we made more significant adversarial changes to preferences, there were cases where the ESDA algorithm failed. This suggests that the assumptions in Theorem 1 are somewhat important for the ESDA algorithm.
> > > > > Regarding your suggestion about comparing the correlation of sibling preferences to non-sibling preferences, we agree that this is significant. In our current model, all family preferences are generated independently. Investigating the correlation could indeed be effective in examining the applicability of the algorithm and understanding the nature of real-world preferences.

---

> > > > > > ### Comment · Reviewer_52vJ · 2024-08-12
> > > > > >
> > > > > > Thank you, your response makes sense, very interesting and I appreciate the discussion.

---

### Official Review · Reviewer_3tiK · 2024-07-11

**Soundness:** 3
**Presentation:** 3
**Contribution:** 3
**Rating:** 7
**Confidence:** 3

**Summary:**

The authors study a variant of the many-to-one matching problem called the stable matching problem with siblings, which generalizes the stable matching problem with couples. In this problem, some families $f \in F$ may have more than one and at most $k$ siblings, ordered by age $(c_1,\dots,c_k)$. Each family $f = (c_1,\dots,c_k)$ expresses a joint linear preference order for daycares, denoted as
$>_f \subseteq D \times D \dots \times D$,

 where $>_{f,j} = (d_1,\dots,d_k)$ represents the $j$th preference of family $f$, and $d_i$ corresponds to the preference for child $c_i$. Note that $>_f$ is an ordered set—a tuple. Each daycare $d \in D$ expresses a linear preference order $>_d \subseteq C$ for a subset of children and a maximum capacity $Q(d)$.

The objective is to find a (stable) matching such that no blocked pair exists. In essence, a blocked pair is a tuple (of edges) $(x_1,\dots,x_\ell)$ and $(y_1,\dots,y_\ell)$ such that swapping $x_i$ with $y_i$ results in a new matching that assigns children to daycares with higher priority for at least one family while not negatively impacting the assignment of any other family or daycare preferences.

A stable matching might not exist for restrictive settings of the problem, as the authors illustrate with a simple example in Appendix B.3. My understanding is that if the preferences form cycles, it becomes impossible to find a feasible solution that satisfies these preference constraints. However, in a daycare market where priorities are generated from a specific distribution, particularly random, the authors demonstrate that the probability of a stable matching existing converges to $1$ as the number of children $n$ approaches infinity.They present algorithms to solve the problem and conduct experiments on synthetic and real-world datasets, demonstrating that they can find feasible solutions in most instances.

**Strengths:**

The paper addresses a challenging and relevant variant of the many-to-one matching problem, the stable matching problem with siblings. The authors provide a comprehensive approach by defining the problem, presenting algorithms to solve it, and conducting thorough experiments on both synthetic and real-world datasets. Their work not only demonstrates the feasibility of finding stable matchings under specific conditions but also highlights the practical applications and implications for real-world daycare allocation scenarios.

**Weaknesses:**

While the paper makes significant contributions, there are some areas that could be improved. The writing is occasionally imprecise, making it challenging to follow the arguments and understand the definitions clearly. In particular, the choice of notation can be confusing (see detailed comments and questions). The structure of the paper is somewhat disorganized, with most of the proofs deferred to the appendix. Considering the strict page limits, this may be reasonable. However, Sections 3 and 4 could be compressed and written more concisely, and some proofs (or at least proof sketches) can be included in the main paper. I have only reviewed the proofs at a high level and have not verified the claims in sufficient detail. Given the strict reviewing timeline, this is the best I can do.

**Questions:**

Line 97: The authors define $ f $ as a function $ f(c) \in F $ which maps $ f: C \rightarrow F $. Then they consider $ f \in F $ as $ f $ being an element in $ F $. This is confusing and imprecise.

Line 98: Using the same symbol for the function and set is incorrect. $ F(f) \subseteq C $ is imprecise; use a different alphabet to denote a function. I think this function should be $ x: f \rightarrow 2^C $ to be precise.

Line 99: $ C(f) = \{c_1,\dots,c_k\} $ should be $ C(f)=(c_1,\dots,c_k) $, since the set is ordered based on age, appropriate notation should be used.

Line 107: Why is the notation used as $ (d_1^*,\dots,d_k^*) $ and $ d_1,\dots,d_k $? What does $ * $ denote here?

Line 113: Using the same function $ \mu $ to map both children to daycares and daycares to children is confusing. Perhaps the authors should reconsider this; an appropriate way would be $ \mu: C \rightarrow D $. Even with the current definition of $ \mu $, using $ \mu(f) $ is very confusing and inaccurate in my opinion, as the function is defined for elements of $C \cup D$ and not a subset of $C \cup D$.

Line 137: Again, $ Ch_d(C') \subseteq C' $ is confusing to parse. It should be $ Ch_d: C' \rightarrow 2^{C'} $ right?

Example 1: What is the capacity of daycares? I am assuming it is $ 1 $.

**Limitations:**

The authors do not discuss limitations and potential negative social impact of their work.

---

> ### Author Rebuttal · Authors · 2024-08-03
>
> Thank you so much for your detailed comments and questions.
>
> Here are the changes based on your suggestions.
>
> Line 97: each child c is associated with one family, denoted as f_c
>
> Line 98: each family f is associated with a set of children, denoted as C_f
>
> Line 99: C_f = (c1, … , ck).
>
> Line 107: In Example 1, we use * to indicate that the same daycare may appear multiple times. For example, (d1, d1) means both children attend the same daycare d1. Here d1* = d2* in the family’s preferences.
>
> Line 113: you are right and we have changed it to \mu: C \rightarrow D.
>
> Line 137: yes, we have changed it as you suggested.
>
> Example 1: The capacity is not explicitly specified because our focus is on explaining family preferences, not on determining the feasibility of the outcome.

---

> > ### Comment · Reviewer_3tiK · 2024-08-08
> > **response to rebuttal**
> >
> > I have read the authors' responses and retain my original assessment.

---

### Official Review · Reviewer_VWTA · 2024-07-13

**Soundness:** 3
**Presentation:** 3
**Contribution:** 3
**Rating:** 6
**Confidence:** 4

**Summary:**

This paper introduces the problem of daycare matching with siblings, an extension of matching with couples. Here, children in families (of size 1 or larger) are matched to daycares. Families have ranked preferences over the tuples of daycares their children end up at (since their preference for one child at one daycare may affect their preference of another child at some daycare), and daycares have preferences over children. In most cases, daycares do not differentiate between children within a family. This is an important problem to solve in Japan, and the authors actually worked with the Japanese daycare matching market in order to produce this work.

Their contributions are: 1) introduce the problem along with notions of rationality/stability/assumptions/etc, 2) propose an extended sorted deferred acceptance algorithm and prove that it will only return stable matchings and will fail to recognize a possible stable matching with probability approaching 1 as the problem grows, and 3) run experiments on their algorithm.

Their model is defined in a pretty standard way according to stable matching literature. The novelty, of course, is the introduction of families generalizing the size of couples. Their stability definition uniquely allows children in the same family to pass along seats to each other, so that a family may use that to their advantage in forming a blocking coalition. They assume that daycares have similar rankings over children and that they are drawn according to the Mallows Model, and that families only have few daycares they are interested in.

The algorithm itself works much like deferred acceptance. First, single children can propose to daycares per usual. Then, families with multiple children begin proposing, presumably according to their full ranking of matching tuples. When a single child is unseated from a daycare, they can simply propose to their next choice. When a family f has an unseated child when family f' is processed, the algorithm attempts again under a new order where f' goes before f. This can cause many iterations.

In the experiments, they use real datasets from Japan as well as larger synthetically-generated datasets. They compare their algorithm to a baseline constraint programming solution, showing that their algorithm returns the same solution faster.

Quick note: diameter is introduced in the main body but only used in the appendix. Perhaps move it to the appendix.

**Strengths:**

Stable matching is a very well-respected area of research, and this seems like a very natural formulation of the problem. It is particularly interesting that the authors are working directly with the market in need and they seem to have been given positive feedback about their work, so this work will almost definitely have a valuable use case. For the most part, the paper is written very well and it is very easy to get a high level understanding of most aspects of the project. Overall, I am very pleased with this paper and would be excited to see it at NeurIPS.

**Weaknesses:**

I am a bit concerned about the literature review provided. I am aware there is much more research that has been conducted on matching markets with complementaries (I am not knowledgeable enough to know what papers would be most useful), and I know there are various papers in this field. However, very few previous works are cited in this paper. It would be great if the authors could clarify the place of their work in the context of current literature and give confidence that this problem or a generalization of it has not already been studied. In fact, this is very important to motivate the paper.

Otherwise, there are a few points in the paper that are unclear. Much of it is very high level and lacks details, which is okay because it writes a narrative, but it comes at a cost of understanding the details of the proofs. More notably, I think the authors didn't spend enough time explaining their algorithm. I found it somewhat vague and I was uncertain about how it worked, and yet it is an integral part of the paper. This definitely needs to be improved.

**Questions:**

1. Line 256: What does the "outcome" refer to? You can't be considering the entire matching - obviously the matching will change when you process a new family.

2. Line 257: "...check whether a family f can be matched to a better tuple..." - What do you mean "can"? What are the conditions for this? Are we saying at the point in $\pi'$ where f is inserted, could it be better matched? Wouldn't they have matched to the preferable tuple in the deferred acceptance methods using $\pi'$? This is very unclear to me.

3. I didn't understand in the algorithm by what mechanism children can transfer seats to siblings. What am I missing?

4. Def 9: Again, I'm likely just missing something here. Domination is defined by the highest priority of family f being prioritized higher than the lowest priority of family f', correct? Doesn't this seem rather likely? In fact, shouldn't there always exist at least one instance of domination in every ordering?

5. Are there any other papers that have studied this problem or a generalization? Where does your work fall in the context of literature?

**Limitations:**

Everything seems adequate.

---

> ### Author Rebuttal · Authors · 2024-08-03
>
> Thank you so much for your detailed comments and questions.
>
> Q1-Q3:
> Once a family f associated with several children is inserted into the market, it may result in the rejection of some children without siblings. If we only check family f’s preferences once, we might overlook a better assignment for it. This could happen if some children pass their seats at current matched daycare to other siblings, while they could move to a different one.
>
> Let’s consider Example 6 in the Appendix. In the first step, child $c_3$ from family $f_2$ is paired with $d_2$. We’ll call this matching $\mu_1$.
>
> Next, we process family $f_1$ with children $c_1$ and $c_2$. Family $f_1$’s top choice is $(d_1, d_2)$. However, $d_2$ is already occupied by $c_3$, who has a higher priority than $c_2$. As a result, family $f_1$ is matched to its second choice $(d_2$, $d_3)$. In this case, $c_3$ is replaced by $c_1$, who has a higher priority. We’ll call this matching $\mu_2$.
>
> Since $\mu_1$ is different from $\mu_2$, the ESDA algorithm does not terminate as the original SDA would. We then check whether family $f_1$ can achieve a better assignment by examining its preferences again. It turns out that family $f_1$ could indeed be matched to its top choice, when child $c_1$ passes his seat at $d_2$ to $c_2$, and child $c_1$ is paired with $d_1$.
> We will clarify this part in the submission to improve its readability.
>
> Q4
> Yes, your understanding of Definition 9 is correct. However, in our proof, our main focus is on Definition 10, which deals with "nesting"—specifically, whether two families, f and f′, dominate each other. For further details, please refer to Example 4.
>
> Q5
> Thank you for your advice and we will add a more detailed literature review later. We have consulted with several experts on two-sided matching and game theory from both economics and computer science. We agree that there is a large body of literature on matching with complementaries, but we found that the most relevant papers on the probability analysis of existence are those by Kojima et al. 2013 and Ashlagi et al. 2014. Recently, several new papers on matching with couples or siblings have been published which focus on maximal preference domains that guarantee a stable matching, or on designing algorithms for certain restrictive preferences. In contrast, we have not modified any preferences in real-life datasets and our goal is to explain why a stable matching exists.
>
> -School choice in Chile. Operations Research, 2022
>
> -Family ties: School assignment with siblings, Theoretical Economics, 2022
>
> -Matching with Externalities, Review of Economic Studies, 2023
>
> -Couples can be tractable: New algorithms and hardness results for the Hospitals / Residents problem with Couples, IJCAI, 2024

---

> > ### Comment · Reviewer_VWTA · 2024-08-12
> >
> > Thank you for your responses. Should this paper get accepted, I would greatly appreciate including more references to matching with complementaries, even if not directly related. For someone like me who is somewhat familiar with that field but doesn't know all the research, this could be very useful for understanding that your work is truly novel. Just stating these works exist and how they differ from yours is important.

---

### Official Review · Reviewer_1G7X · 2024-07-15

**Soundness:** 4
**Presentation:** 4
**Contribution:** 3
**Rating:** 6
**Confidence:** 3

**Summary:**

This paper studies the existence of a stable daycare-children matching in the presence of siblings from the same families with same preferences over the daycares. The authors particularly study the case when the daycares have similar preferences over the set of children, and the market size is large. They propose a variant of the Sorted Deferred Acceptance algorithm to compute the stable matchings.

**Strengths:**

1. The problem is well motivated by the real-world observation that stable matchings exist in the markets as opposed to what the theory suggests. This observation allowed the authors to make necessary adjustments to the assumptions that are sufficient for the theory to work out.
2. The authors take a systematic approach to the problem. They first define a new notion of stability that takes siblings into consideration and show that stable matchings may not exist in the presence of siblings and that the previous algorithms do not work for this new notion of stability. They then consider a specific random daycare market, mention the drawbacks of the existing methods of computing stable matchings, and then prove that a modification to the existing algorithm can find stable matchings with the new definition of stability.
3. The results by themselves are quite interesting; that stable matchings exist even in the presence of siblings with complementaries.
4. The analogy is drawn between the related work in stable matchings with couples and stable matchings with siblings

**Weaknesses:**

1. The assumption that day cares have similar priorities over children is slightly unrealistic.
2. The random daycare market for which the results are derived is somewhat restrictive.

**Questions:**

1. Can the authors give some evidence for whether daycare centers do have similar priorities over children or that the daycare centers use a priority scoring function? Is this just an assumption made by the authors to simplify the computation of stable matchings, or is there evidence supporting this assumption?
2. Is the priority scoring function the same across all the daycares?
3. Can you repeat the algorithm multiple times until you get a successful matching, if the dispersion parameter is large?

**Limitations:**

Limitations sufficiently addressed by the authors.

---

> ### Author Rebuttal · Authors · 2024-08-03
>
> Thank you so much for your detailed comments and questions.
>
> Q1 and Q2:
> We are actively collaborating with multiple municipalities in Japan. In current daycare markets, each municipality establishes a unique and complex priority scoring system that is publicly accessible. Typically, children from low-income or single-parent households, or those with guardians facing health issues or disabilities, are given higher priority. Ties in scores are often resolved using additional rules.
>
> Once priority scores are calculated, they are applied by all daycare centers, with minor adjustments according to each center's regulations. For example, a child with a sibling already enrolled at a daycare may receive additional points for that particular center.
>
> Overall, the priority score for each child tends to be consistent across daycare centers, which is a notable feature of the Japanese daycare system.
>
> Q3:
> Thank you for your suggestion. However, the proposed solution may not work well because there is a high likelihood of rejection cycles (see Appendix C for details), even if we apply the algorithm with different permutations of families. Our next goal is to design a more robust algorithm that still performs well even with a large dispersion parameter.

---

### Author Rebuttal · Authors · 2024-08-05

We appreciate the efforts of all the reviewers and their valuable feedback. We are pleased to address their suggestions, which are detailed in our individual responses.

1. One notable feature we observed in the Japanese daycare matching market is the similarity of priority scores for each child across all daycares.
2. We acknowledge the reviewers' feedback regarding the imprecise definitions and have revised them accordingly.
3. We will include a more detailed literature review on recent developments in matching with complementarities. Please note that the most relevant papers on the probability analysis of existence are those by Kojima et al. 2013 and Ashlagi et al. 2014.

---

### Decision · Program_Chairs · 2024-09-25

**Decision:**

Reject

**Comment:**

The authors study a generalization of the stable matching problem (stable matching with siblings). All of the reviewers agreed that the theoretical and empirical contributions are interesting and the work is generally well done. However, the core audience for this work is unlikely to find itself at NeurIPS, and the paper's results, while solid, are not so groundbreaking or broad as to easily transcend areas.

As such, this is a borderline decision, weighing the strength of the results against the recognition, support, and follow-up work they would get from publishing outside of the core discipline. Upon discussion with the senior area chair, it is our opinion that this work's focus is relatively narrow and it deserves to be published in a more AGT focused venue, be it EC, SAGT, or other related conferences, where it's likely to be well received.